# Cross-modal Active Complementary Learning with Self-refining Correspondence

**Yang Qin**[1], **Yuan Sun**[1], **Dezhong Peng**[1,3,4], **Joey Tianyi Zhou**[2],
**Xi Peng**[1], **Peng Hu**[1]*

[1] College of Computer Science, Sichuan University, Chengdu, China.
[2] Centre for Frontier AI Research (CFAR) and Institute of High
Performance Computing (IHPC), A*STAR, Singapore.
[3] Chengdu Ruibei Yingte Information Technology Co., Ltd, Chengdu, China.
[4] Sichuan Zhiqian Technology Co., Ltd, Chengdu, China.
`{qinyang.gm,joey.tianyi.zhou,pengx.gm,penghu.ml}@gmail.com`,
`sunyuan_work@163.com, pengdz@scu.edu.cn`

## Abstract

Recently, image-text matching has attracted more and more attention from academia and industry, which is fundamental to understanding the latent correspondence across visual and textual modalities. However, most existing methods implicitly assume the training pairs are well-aligned while ignoring the ubiquitous annotation noise, *a.k.a* noisy correspondence (NC), thereby inevitably leading to a performance drop. Although some methods attempt to address such noise, they still face two challenging problems: excessive memorizing/overfitting and unreliable correction for NC, especially under high noise. To address the two problems, we propose a generalized Cross-modal Robust Complementary Learning framework (CRCL), which benefits from a novel Active Complementary Loss (ACL) and an efficient Self-refining Correspondence Correction (SCC) to improve the robustness of existing methods. Specifically, ACL exploits active and complementary learning losses to reduce the risk of providing erroneous supervision, leading to theoretically and experimentally demonstrated robustness against NC. SCC utilizes multiple self-refining processes with momentum correction to enlarge the receptive field for correcting correspondences, thereby alleviating error accumulation and achieving accurate and stable corrections. We carry out extensive experiments on three image-text benchmarks, *i.e.*, Flickr30K, MS-COCO, and CC152K, to verify the superior robustness of our CRCL against synthetic and real-world noisy correspondences. Code is available at `https://github.com/QinYang79/CRCL`.

## 1 Introduction

Image-text matching aims to search the most relevant samples across different modalities, which is fundamental for most cross-modal tasks [1, 2, 3, 4, 5, 6]. The core of image-text matching is how to accurately measure the similarity between distinct modalities, however, which is challenging due to the visual-textual discrepancy. To tackle the challenge, numerous deep methods are presented to learn the visual-semantic associations of image-text pairs and achieve remarkable progress, thanks to the powerful representation ability of Deep Neural Networks (DNNs) and some well-designed similarity inference architectures [7, 8, 4, 9]. They could be roughly divided into two groups, *i.e.*, global-level methods [7, 9] and local-level methods [8, 4], which aim at learning the image-to-sentence and region-to-word correlation to infer the cross-modal similarity, respectively. Although these methods

---

*Corresponding author.

37th Conference on Neural Information Processing Systems (NeurIPS 2023).

achieved promising matching performance, most of them implicitly require large-scale well-aligned data for training, which is expensive or even impossible to collect due to ubiquitous noise in real-world scenarios [10, 11]. Therefore, there is inevitably imperfect alignment luring in the data, *i.e.*, noisy correspondence (NC) [12], resulting in inferior performance.

To handle the NC problem, some prior arts are presented to alleviate the adverse impact of NC in various tasks, *e.g.*, partially view-aligned clustering [13, 14, 15, 16], video-text retrieval [17], visible-infrared person re-identification [18], and image-text matching [12, 19, 20]. Specifically, inspired by learning with noisy labels [21, 22], some works [12, 23, 24] are proposed to alleviate the negative impact brought by NC. These works attempt to leverage the memorization effect of DNNs [25] to gradually distinguish the noisy image-text pairs for robust learning in a co-teaching manner. Furthermore, the predicted soft correspondence is used to recast a soft margin to replace the scalar margin of triplet ranking loss [7], which helps to avoid misleading the model by mismatched pairs. However, the soft-margin ranking loss is experimentally found to provide only limited robustness against NC, especially under high noise (as shown in Table 1), due to unstable division based on inaccurate predictions. In contrast, some works [19, 20] aim to enhance the robustness of cross-modal methods against NC by starting with a robust loss function, *i.e.*, avoiding over-amplification of wrong supervision information to reduce misleading risk. However, the lack of explicitly mitigating the effect of easily separable noise makes them hard to further improve the performance.

To address the aforementioned issues, we propose a generalized robust framework, dubbed CRCL, for learning with noisy correspondences. CRCL could be easily migrated into existing image-text matching methods to enhance their robustness against NC. Our framework introduces a novel Active Complementary Loss (ACL) that applies active and complementary learning to mutually boost robustness against NC. Specifically, we present a robust complementary learning loss that employs complementary pairs, such as "input image-text pairs are irrelevant", to conduct indirect cross-modal learning with exponential normalization. Due to the low likelihood of selecting incorrect complementary pairs, robust complementary learning could reduce the risk of providing incorrect supervision and smooth the losses, thus embracing robustness against NC. However, the robust loss will face the underfitting problem, leading to suboptimal performance. To overcome this issue, a weighted Active learning loss is proposed to enforce the model focus on more reliable positive pairs in addition to only complementary pairs. In addition, we propose a Self-refining Correspondence Correction paradigm (SCC) to obtain stable and accurate correspondence correction. SCC utilizes Momentum Correction (MC) to aggregate historical predictions for stable and accurate correspondence corrections. By combining multiple Self-Refining processes (SR) throughout the entire training process, we alleviate over-memorization for NCs. In summary, the key contributions and innovations of this work are as follows:

- We propose a generalized Cross-modal Robust Contrastive Learning framework (CRCL) to address a pressing and widely-exist problem in image-text matching, *i.e.*, noisy correspondence. CRCL empowers existing methods with strong robustness through the perspectives of robust loss and correction techniques.

- A novel Active Complementary Loss (ACL) is presented to balance active and complementary learning, mutually enhancing robustness against NC while encapsulating correct cross-modal associations in the latent common space.

- We design an effective Self-refining Correspondence Correction paradigm (SCC) to achieve accurate and stable soft correspondences, which enables the prediction-based corrections to perceive larger fields and self-refine from the historically learned knowledge.

- Extensive experiments verify the effectiveness and superiority of our framework on three benchmark image-text datasets: Flickr30K, MS-COCO, and CC152K. Additionally, comprehensive ablation studies and insightful analyses demonstrate the reliability and practicability of the proposed CRCL.

## 2 The Proposed Method

### 2.1 Preliminaries and Problem Statement

To be specific, we first provide some definitions of instance-level image-text matching so as to conveniently study noisy correspondence. Let $\mathcal{D} = \{\mathcal{I}, \mathcal{T}, \mathcal{Y}\}$ be an image-text dataset, where $\mathcal{I} =$

$\{I_i\}_{i=1}^{N}$ and $\mathcal{T} = \{T_i\}_{i=1}^{N}$ are the training image and text set with size of $N$. The correspondence label space is defined as $\mathcal{Y} = \{y_{ij} | i = 1, \cdots, N; j = 1, \cdots, N\}$, where $y_{ij}$ represents the correspondence of pair $(I_i, T_j)$, *i.e.*, if $I_i$ and $T_j$ are matched (*i.e.*, positive pair), $y_{ij} = 1$ otherwise $y_{ij} = 0$. We assume each pair with the same indices has matched correspondence, *i.e.*, $y_{ii} = 1, i = 1, \cdots, N$. However, due to the ubiquitous noise during data collection, some negative pairs are mismatched as positives, *a.k.a* noisy correspondence (NC) [12, 19], which would introduce wrong supervisory information and misguide model training, leading to performance degradation. Mathematically, we define NC as shown in Definition 1.

**Definition 1.** *Due to the existence of NC, the learner only has access to the noisy training data $(\mathcal{D}_\eta)$, instead of clean data $(\mathcal{D})$. Thus, the correspondence label for pair $(I_i, T_j)$ is reconsidered as*

$$\tilde{y}_{ij} = \begin{cases} y_{ij} & \text{with probability } (1 - \eta_{ij}), \\ 1 - y_{ik} & \text{with probability } \bar{\eta}_{ik}, \forall k \neq j. \end{cases} \quad (1)$$

*For all pairs, conditioned on that if $i = j$ then $y_{ij} = 1$ else $y_{ij} = 0$, we have $\sum_{j \neq k} \bar{\eta}_{ik} = \eta_{ij}$. Similar to the definitions of noisy labels [26], we assume that NC is uniform, i.e., $\eta_{ij} = \eta$ and $\bar{\eta}_{ik} = \frac{\eta}{N-1}, \forall k \neq j$, where $\eta$ is a constant to represent the noise rate.*

The key to learning with noisy correspondence is to alleviate the misguiding impact and rectify the mismatched pairs. One direct solution is to enhance the robustness of loss function $\mathcal{L}$ against noisy pairs, which can help prevent overfitting on mismatched pairs. The second aims at using the memorization effect of DNNs [25] to discriminate the mismatched pairs, thus removing unreliable supervision from the training data.

For image-text matching, images and texts are first projected into a shared representation space by two modality-specific networks, denoted as $v$ and $g$, respectively. The cross-modal similarity of a given pair $(I_i, T_j)$ is then computed as $S(v(I_i), g(T_j))$, where $S(*)$ could be the cosine function [7, 9] or a relevance inference module [8, 4]. For brevity, $S(v(I_i), g(T_j))$ is denoted as $S(I_i, T_j)$ or $S_{ij}$ in the following. The computed similarities could be considered as supporting evidence for retrieved results. Thus, the learning objective of image-text matching is to maximize the cross-modal similarities of positive pairs while minimizing those of negatives in the latent common space, which is commonly achieved by using contrastive learning [7, 8, 4].

The widely-used triplet ranking loss [7] has shown excellent performance in cross-modal contrastive learning tasks [7, 27]. However, recent research [19] has demonstrated that this loss function fails to perform well in image-text data with NCs, especially when using the hardest negative samples as comparison items. To address this issue, some works proposed an adaptive soft margin approach to improve the robustness of the ranking loss [12, 24, 23], which is defined as follows:

$$\mathcal{L}_{soft}(I_i, T_i) = \left[\hat{\alpha}_i - S(I_i, T_i) + S\left(I_i, \hat{T}_h\right)\right]_+ + \left[\hat{\alpha}_i - S(I_i, T_i) + S\left(\hat{I}_h, T_i\right)\right]_+, \quad (2)$$

where $\hat{T}_h$ and $\hat{I}_h$ denote the hardest cross-modal samples in a mini-batch, $\hat{\alpha}_i$ is a soft margin adaptively computed by $\hat{\alpha}_i = \frac{m^{\hat{y}_{ii}} - 1}{m - 1}\alpha$, $\alpha$ is a constant margin, $m$ is a curve parameter, and $\hat{y}_{ii}$ is the rectified correspondence between $I_i$ and $T_i$. However, this approach has two disadvantages: 1) The margin setting $\alpha$ may not be consistent with the empirical setting under NC scenarios. 2) The inaccurate prediction $\hat{y}_{ii}$ can still easily produce the risk of misleading gradient, which can cause trivial solutions to fail, especially in the case of high noise (*e.g.*, the results of NCR [12] and BiCro [23] on Flickr30K with 80% noise). To overcome these limitations, we propose a novel Active Complementary Loss (ACL) under the risk minimization theory [28, 26] to provide noise tolerance for noisy pairs while ensuring discriminative learning.

## 2.2 Active Complementary Loss

For the image-text dataset $\mathcal{D}$, our goal is to learn a cross-modal model $(\mathcal{M})$ that can discriminatively identify the positive (matched) and negative (unmatched) pairs well for retrieval, which is intuitively equivalent to maximizing bidirectional matching probabilities of positives. The bidirectional matching probabilities for pair $(I_i, T_j)$ are defined as:

$$p_{ij}^{\circ} = f(I_i, T_j) = \frac{e^{(S_{ij}/\tau)}}{\sum_{l=1}^{N} e^{(S_{il}/\tau)}}, \quad p_{ij}^{\diamond} = f(T_j, I_i) = \frac{e^{(S_{ij}/\tau)}}{\sum_{l=1}^{N} e^{(S_{lj}/\tau)}}, \quad (3)$$

where $\tau$ is a temperature parameter [29, 30, 31], $f$ is regarded as the cross-modal decision function, and "$\circ, \diamond$" means the two retrieval directions, i.e., image-to-text and text-to-image, respectively. For any loss function $\mathcal{L}$, the matching risk of $f$ for image-text matching can be defined as

$$R_{\mathcal{L}}(f) = \mathbb{E}_{(I_i, y_{i\cdot}) \sim \mathcal{D}} \left[ \mathcal{L}(f(I_i, T_\cdot), y_{i\cdot}) \right] + \mathbb{E}_{(T_i, y_{\cdot i}) \sim \mathcal{D}} \left[ \mathcal{L}(f(T_i, I_\cdot), y_{\cdot i}) \right], \tag{4}$$

where $\mathbb{E}[\cdot]$ represents the expectation operator. Considering noisy correspondences, the risk of $f$ in noisy data $\mathcal{D}_\eta$ could be formulated as follows:

$$R_{\mathcal{L}}^{\eta}(f) = \mathbb{E}_{(I_i, \tilde{y}_{i\cdot}) \sim \mathcal{D}_\eta} \left[ \mathcal{L}(f(I_i, T_\cdot), \tilde{y}_{i\cdot}) \right] + \mathbb{E}_{(T_i, \tilde{y}_{\cdot i}) \sim \mathcal{D}_\eta} \left[ \mathcal{L}(f(T_i, I_\cdot), \tilde{y}_{\cdot i}) \right], \tag{5}$$

where $\tilde{y}_{ij}$ is the noisy correspondence label as shown in Definition 1. Thus, the cross-modal learning objective is to learn a model $\mathcal{M}_{\mathcal{L}}^*$ with a global minimizer $f_\eta^*$ of $R_{\mathcal{L}}^{\eta}(f)$. To achieve robustness, $f_\eta^*$ should be also the global minimizer of $R_{\mathcal{L}}(f)$ on the noise-free data.

Inspired by complementary contrastive learning [20], we propose to optimize the matching probabilities of all negative pairs for learning with noisy data indirectly, thereby avoiding fast overfitting to NC. Simultaneously, to further improve the noise tolerance, we introduce an exponential normalization to smooth the complementary loss. Hence, the robust complementary loss for pair $(I_i, T_i)$ could be formulated as:

$$\begin{aligned}
\mathcal{L}_r(I_i, T_i, q) &= \mathcal{L}_r^{\circ}(I_i, T_i, q) + \mathcal{L}_r^{\diamond}(T_i, I_i, q) \\
&= \sum_{j \neq i}^{N} \tan(p_{ij}^{\circ}) \Big/ \Big( \sum_{k=1}^{N} \tan(p_{ik}^{\circ}) \Big)^q + \sum_{j \neq i}^{N} \tan(p_{ji}^{\diamond}) \Big/ \Big( \sum_{k=1}^{N} \tan(p_{ki}^{\diamond}) \Big)^q,
\end{aligned} \tag{6}$$

where $\tan(\cdot)$ is the tan function and $q \in [0, 1]$ is a regulatory factor. Theoretically, for any input $(I_i, T_i)$ under noise rate $\eta \leq \frac{N-1}{N}$, we can show (see proofs in supplementary material)

$$C \leq R_{\mathcal{L}_r}(f^*) - R_{\mathcal{L}_r}(f_\eta^*) \leq 0, \tag{7}$$

where $C = 2\eta(A_{\min}^{(1-q)} - A_{\max}^{(1-q)})/(1 - \frac{N\eta}{N-1}) \leq 0$. $C$ increases as $q$ increases and when $q = 1$, $C$ takes the maximum value 0. $A_{\min}$ and $A_{\max}$ are the maximum and minimum values of $\sum_{j=1}^{N} \tan(p_{ij})$ under the condition $\sum_{j=1}^{N} p_{ij} = 1$, where $1 < A_{\min} < A_{\max}$, and $0 \leq p_{ij} \leq 1$ ($p_{ij} = p_{ij}^{\circ}$ or $p_{ij}^{\diamond}$). $f^*$ and $f_\eta^*$ are the global minimizers of $R_{\mathcal{L}_r}(f)$ and $R_{\mathcal{L}_r}^{\eta}(f)$, respectively.

**Analysis:** The larger the $q$ is, $C \to 0$, the tighter the bound of Equation (7) is. When $q = 0$, $\mathcal{L}_r$ is a standard complementary contrastive loss [20]. In the extreme case of $q = 1$, $R_{\mathcal{L}_r}(f^*) = R_{\mathcal{L}_r}(f_\eta^*)$, i.e., $\mathcal{L}_r$ is noise tolerant since $f_\eta^*$ and $f^*$ are simultaneously the minimizers of $R_{\mathcal{L}_r}(f)$ (It can also be obtained from Lemma 1 that $\mathcal{L}_r$ is robust under $q = 1$). Put another way, as $q$ approaches 1, the optimum $f_\eta^*$ of the noisy risk will be close to $f^*$ on the clean data more likely, which implies noise tolerance.

Like most robust learning methods [32, 33], we should focus more on reliable data, while less on unreliable data. In other words, a smaller value of $q$ is used for more convincing pairs (with larger $\hat{y}_{ii}$), while a larger value of $q$ is for less convincing pairs (with smaller $\hat{y}_{ii}$). Thus, we could empirically utilize the soft corrected label $\hat{y}_{ii}$ to recast $q$ like the soft margin used in NCR [12], i.e., $q = 1 - \hat{y}_{ii}$. However, indirect learning will face the underfitting problem, resulting in suboptimal/insufficient performance. To address this issue, we introduce a weighted active learning loss $\mathcal{L}_d$ to make the model pay more attention to positive/matched pairs, i.e., $\mathcal{L}_d(I_i, T_i, \tilde{y}_{ii}) = -\tilde{y}_{ii} (\log p_{ii}^{\circ} + \log p_{ii}^{\diamond})$. This positive learning will mine discrimination from direct supervision, which complements the complementary learning loss. By combining active and complementary learning losses, our active complementary loss is defined as:

$$\mathcal{L}_{acl}(I_i, T_i, \hat{y}_{ii}) = \mathcal{L}_d(I_i, T_i, \hat{y}_{ii}) + \lambda \mathcal{L}_r(I_i, T_i, 1 - \hat{y}_{ii}), \tag{8}$$

where $\lambda$ is a scale factor to prevent $\mathcal{L}_d$ from dominating the cross-modal training and quickly overfitting NC. As shown in Equation (8), when $(I_i, T_i)$ is a noisy pair and $\hat{y}_{ii}$ ideally approaches 0, the loss emphasizes robust complementary learning, thus mitigating overfitting to NC. Conversely, when $(I_i, T_i)$ is a clean pair and $\hat{y}_{ii}$ ideally approaches 1, the loss focuses on discriminative learning, thereby facilitating the accurate acquisition of visual-semantic associations. However, due to computational resource constraints, we cannot use the entire training set to perform cross-modal learning. Therefore,

we relax $N$ to the size $K$ of the mini-batch $\mathbf{x}$ by Monte Carlo sampling. Without loss of generality, the final loss for cross-modal learning is given by:

$$\mathcal{L}_{acl}(\mathbf{x}) = \frac{1}{K} \sum_{i=1}^{K} \mathcal{L}_{acl}(I_i, T_i, \hat{y}_{ii}). \tag{9}$$

**Lemma 1.** *In an instance-level cross-modal matching problem, under uniform NC with noise rate $\eta \leq \frac{N-1}{N}$, when $q = 1$, $\mathcal{L}_r$ is noise tolerant.*

*Proof.* The proofs of Equation (7) and Lemma 1 can be found in the supplementary material. $\square$

### 2.3 Self-refining Correspondence Correction

Another key to solving NC is how to obtain accurate correspondence estimations so as to reduce the adverse effects of NC. To this end, we propose an effective Self-refining Correspondence Correction paradigm (SCC). SCC leverages Momentum Correction (MC) to aggregate historical predictions, providing stable and accurate correspondence estimations while alleviating the over-memorization to NC. To eliminate the error accumulation against NC, we combine multiple independent Self-Refining (SR) in the entire training process. Specifically, the MC for the correspondence of $(I_i, T_i)$ at the $t$-th epoch is defined as follows:

$$y_{ii}^t = \beta y_{ii}^{t-1} + (1 - \beta)\hat{p}^t(I_i, T_i), \tag{10}$$

where $\beta \in (0, 1)$ represents the momentum coefficient, $\hat{p}^t(I_i, T_i) = (p_{ii}^{\circ} + p_{ii}^{\diamond})/2$ denotes the average matching probability at the $t$-th epoch. Through adequate cross-modal training, our CRCL will obtain a more stable and accurate soft correspondence label by smoothly evolving and expanding the receptive field of correction based on historical predictions with MC. Notably, as training progresses, some pairs would always be incorrectly distinguished as clean or noisy ones, resulting in the error accumulation of the estimated labels (see Figure 1c). Additionally, even though the updates performed by MC help reduce the negative influence of NC, the initial correspondence label $\left(y_{ii}^0\right)$ still greatly affects the quality of subsequent smooth corrections. In other words, providing more accurate initial correspondences weakens the DNN's memorization against NC, thereby reducing the risk of error accumulation.

---

**Algorithm 1:** The pseudo-code of CRCL

---

**Input:** A noisy training dataset $\mathcal{D}_\eta$, image-text matching model $\mathcal{M}(\Theta)$;
**Initialize:** $\Theta$;
**for** $e^j$ *in* $[e_1, e_2, \cdots, e_m]$ **do**
    **for** $t$ *in* $[1, 2, \cdots, e^j]$ **do**
        **for** $\mathbf{x}$ *in batches* **do**
            Obtain the bidirectional matching probabilities of $\mathbf{x}$ with Equation (3);
            Update the correspondence labels with Equation (11);
            Obtain the corrected labels with Equation (12);
            Compute the overall loss $\mathcal{L}_{acl}(\mathbf{x})$ with Equation (9);
            $\Theta = \text{Optimizer}(\Theta, \mathcal{L}_{acl}(\mathbf{x}))$;
        **end**
    **end**
    Re-initialize $\Theta$;
**end**
**Output:** The learned parameters $\hat{\Theta}$;

---

To achieve accurate initial correspondence estimations, SCC refines updated correspondence labels historically in epochs using MC through multiple concatenated SR pieces during the entire training process. Subsequent SR pieces could gradually aggregate the learned correspondence knowledge from previous pieces, thus improving the quality of estimated correspondences progressively. Furthermore, each SR piece is trained from scratch, which aims to clear accumulated error/noise that has been memorized, thus providing more accurate correspondence predictions for subsequent training. Mathematically, SCC consists of multiple SR pieces ($[e_1, \cdots, e_m]$) based on MC, where each piece

undergoes robust learning for $e_j$ ($j \in \{1, \cdots, m\}$) epochs. Thus, during the $j$-th SR training, the estimated soft label of the $i$-th pair at $t$-th epoch is reconsidered as follows:

$$y_{ii}^{(j,t)} = \begin{cases} y_{ii}^{(j-1,e_{j-1})}, & \text{if } t \leq e_f, \\ \hat{p}^{(j,t-1)}(I_i, T_i), & \text{if } j = 1 \text{ and } t = (e_f + 1), \\ \beta y_{ii}^{(j,t-1)} + (1-\beta)\hat{p}^{(j,t-1)}(I_i, T_i), & \text{otherwise,} \end{cases} \quad (11)$$

where $\hat{p}^{(j,t-1)}(*)$ is the average matching probability of pair $(I_i, T_i)$ at the $(t-1)$-th epoch during the $j$-th training piece, $e_f$ denotes the number of epochs to freeze the correspondence label, preventing insufficient model training in the early stage from affecting the correction quality. In our experiments, we set all initial labels to 1, assuming that all training pairs are matched at the beginning of the first SR piece. In practice, we assign the label of the confident noisy pair as 0 to reduce the risk of producing erroneous supervision information. Therefore, the final corrected correspondence label used for $\mathcal{L}_{acl}$ is defined as:

$$\hat{y}_{ii} = \begin{cases} 0, & \text{if } y_{ii}^{(j,t)} < \epsilon, \\ y_{ii}^{(j,t)}, & \text{otherwise,} \end{cases} \quad (12)$$

where $\epsilon = 0.1$ is a fixed threshold used to filter the confident noisy pairs in experiments.

## 3 Experiments

In this section, comprehensive experiments are conducted on three widely used benchmarks to demonstrate the robustness and effectiveness of our CRCL under multiple scenarios, including synthetic noise, real-world noise, and well-annotated correspondence.

### 3.1 Datasets and Protocols

**Datasets:** For an extensive evaluation, we use three benchmark datasets (*i.e.*, Flickr30K [34], MS-COCO [35] and CC152K [12]) in our experiments. More specifically, Flickr30K is a widely-used image-text dataset collected from the Flickr website, which comprises 31,000 images and each one has 5 corresponding textual descriptions. Following [36], 30,000 images are employed for training, 1,000 images for validation, and 1,000 images for testing in our experiments. MS-COCO is a large-scale image-text dataset, which has 123,287 images, and 5 captions are given to describe each image. We follow the split of [36, 8] to carry out our experiments, *i.e.*, 5000 validation images, 5000 test images, and the rest for training. CC152K is a subset of Conceptual Captions (CC) [10] collected in the real world, which is selected by [12]. Due to the absence of manual annotation, there are about $3\% \sim 20\%$ incorrect correspondences in CC, *i.e.*, real-world noisy correspondences. CC152K contains 150,000 image-text pairs for training, 1,000 pairs for validation, and 1,000 pairs for testing.

**Evaluation Protocols:** Recall at K (R@K=1, 5, and 10) is used to measure the performance of bidirectional retrievals, which is defined as the proportion of the queries with the correct item in the top K retrieved results. Besides, flowing [19], we also take the sum of all Recalls to evaluate the overall performance, *i.e.*, rSum.

### 3.2 Implementation Details

Our CRCL is a generalized robust framework that could extend existing methods to confront noisy correspondences. To demonstrate the effectiveness and robustness of CRCL, we extend two representative methods, *i.e.*, VSE$\infty$[9] and SGRAF (SGR and SAF) [4], to perform robust image-text matching, respectively. Specifically, the shared hyper-parameters are set as the same as the original works [4, 9], *e.g.*, the batch size is 128, the word embedding size is 300, and the joint embedding dimensionality is 1,024. More specific hyper-parameters and implementation details are given in our supplementary material due to the space limitation.

### 3.3 Comparison with State-of-the-Arts

In this section, we evaluate our CRCL by comparing it with 7 state-of-the-art methods on three benchmarks, *i.e.*, SCAN (ECCV'18) [8], SGRAF (SGR and SAF, AAAI'21) [4], VSE$\infty$ (CVPR'21) [9], NCR (NeurIPS'21) [12], DECL (ACM MM'22) [19], BiCro (CVPR'23) [23] and MSCN

(CVPR'23) [24]. For a fair comparison, all tested approaches adopt the same visual features (BUTD features) [8] and textual backbone Bi-GRU [37]. To comprehensively investigate the robustness of our method, we artificially inject synthetic false correspondence of different ratios by proportionally shuffling the captions on Flickr30K and MS-COCO like [12], *i.e.*, 20%, 40%, 60%, and 80% noise rates. In addition to synthetic noise, we also evaluate the robustness of tested methods against the real-world noisy correspondences on CC152K. Due to the space limitation, we only provide the results on MS-COCO 5K under well-annotated correspondences in Table 2. For fairness, like [19, 23], the ensemble results of CRCL-SGRAF are reported in the paper. More extensive comparison results are provided in the supplementary material to fully demonstrate the superiority of CRCL.

Table 1: Performance comparison (R@K(%) and rSum) of image-text matching on Flickr30K and MS-COCO 1K. The highest scores are shown in **bold**. '*' means robust methods.

| | | Flickr30K | | | | | | | MS-COCO 1K | | | | | | |
| | | Image → Text | | | Text → Image | | | | Image → Text | | | Text → Image | | | |
| Noise | Methods | R@1 | R@5 | R@10 | R@1 | R@5 | R@10 | rSum | R@1 | R@5 | R@10 | R@1 | R@5 | R@10 | rSum |
|---|---|---|---|---|---|---|---|---|---|---|---|---|---|---|---|
| 20% | SCAN | 56.4 | 81.7 | 89.3 | 34.2 | 65.1 | 75.6 | 402.3 | 28.9 | 64.5 | 79.5 | 20.6 | 55.6 | 73.5 | 322.6 |
| | SAF | 51.8 | 79.5 | 88.3 | 38.1 | 66.8 | 76.6 | 401.1 | 41.0 | 78.4 | 89.4 | 38.2 | 74.0 | 85.5 | 406.5 |
| | SGR | 61.2 | 84.3 | 91.5 | 44.5 | 72.1 | 80.2 | 433.8 | 49.1 | 83.8 | 92.7 | 42.5 | 77.7 | 88.2 | 434.0 |
| | VSE∞ | 69.0 | 89.2 | 94.8 | 48.8 | 76.3 | 83.8 | 461.9 | 73.5 | 93.3 | 97.0 | 57.4 | 86.5 | 92.8 | 500.5 |
| | NCR* | 76.7 | 93.9 | 96.9 | 57.5 | 82.8 | 89.2 | 497.0 | 77.0 | 95.6 | 98.1 | 61.5 | 89.3 | 95.1 | 516.6 |
| | DECL* | 75.6 | 93.8 | 97.4 | 58.5 | 82.9 | 89.4 | 497.6 | 77.1 | 95.9 | 98.4 | 61.6 | 89.1 | 95.2 | 517.3 |
| | BiCro* | **78.1** | 94.4 | 97.5 | 60.4 | 84.4 | 89.9 | 504.7 | 78.8 | 96.1 | 98.6 | 63.7 | 90.3 | 95.7 | 523.2 |
| | MSCN* | 77.4 | 94.9 | 97.6 | 59.6 | 83.2 | 89.2 | 501.9 | 78.1 | **97.2** | **98.8** | 64.3 | 90.4 | 95.8 | 524.6 |
| | **CRCL*** | 77.9 | **95.4** | **98.3** | **60.9** | **84.7** | **90.6** | **507.8** | **79.6** | 96.1 | 98.7 | **64.7** | **90.6** | **95.9** | **525.6** |
| 40% | SCAN | 29.9 | 60.5 | 72.5 | 16.4 | 38.5 | 48.6 | 266.4 | 30.1 | 65.2 | 79.2 | 18.9 | 51.1 | 69.9 | 314.4 |
| | SAF | 34.3 | 65.6 | 78.4 | 30.1 | 58.0 | 68.5 | 334.9 | 36.0 | 74.4 | 87.0 | 33.7 | 69.4 | 82.5 | 383.0 |
| | SGR | 47.2 | 76.4 | 83.2 | 34.5 | 60.3 | 70.5 | 372.1 | 43.9 | 78.3 | 89.3 | 37.0 | 72.8 | 85.1 | 406.4 |
| | VSE∞ | 30.2 | 58.3 | 70.2 | 22.3 | 49.6 | 62.7 | 293.3 | 53.3 | 84.3 | 92.1 | 31.4 | 63.8 | 75.0 | 399.9 |
| | NCR* | 75.3 | 92.1 | 95.2 | 56.2 | 80.6 | 87.4 | 486.8 | 76.5 | 95.0 | 98.2 | 60.7 | 88.5 | 95.0 | 513.9 |
| | DECL* | 72.5 | 93.1 | 97.0 | 55.8 | 81.2 | 88.1 | 487.7 | 77.1 | 95.7 | **98.3** | 61.5 | 89.2 | 95.3 | 517.1 |
| | BiCro* | 74.6 | 92.7 | 96.2 | 55.5 | 81.1 | 87.4 | 487.5 | 77.0 | **95.9** | **98.3** | 61.8 | 89.2 | 94.9 | 517.1 |
| | MSCN* | 74.4 | 94.4 | 96.9 | 57.2 | 81.7 | 87.6 | 492.2 | 74.8 | 94.9 | 98.0 | 60.3 | 88.5 | 94.4 | 510.9 |
| | **CRCL*** | **77.8** | **95.2** | **98.0** | **60.0** | **84.0** | **90.2** | **505.2** | **78.2** | 95.7 | **98.3** | **63.3** | **90.3** | **95.7** | **521.5** |
| 60% | SCAN | 16.9 | 39.3 | 53.9 | 2.8 | 7.4 | 11.4 | 131.7 | 27.8 | 59.8 | 74.8 | 16.8 | 47.8 | 66.4 | 293.4 |
| | SAF | 28.3 | 54.5 | 67.5 | 22.1 | 47.3 | 59.0 | 278.7 | 28.2 | 63.9 | 79.4 | 31.1 | 65.6 | 80.5 | 348.7 |
| | SGR | 28.7 | 58.0 | 71.0 | 23.8 | 49.5 | 60.7 | 291.7 | 37.6 | 73.3 | 86.3 | 33.8 | 68.6 | 81.7 | 381.3 |
| | VSE∞ | 18.0 | 44.0 | 55.7 | 15.1 | 38.5 | 51.8 | 223.1 | 33.4 | 64.8 | 79.1 | 26.0 | 60.1 | 76.3 | 339.7 |
| | NCR* | 68.7 | 89.9 | 95.5 | 52.0 | 77.6 | 84.9 | 468.6 | 72.7 | 94.0 | 97.6 | 57.9 | 87.0 | 94.1 | 503.3 |
| | DECL* | 69.4 | 89.4 | 95.2 | 52.6 | 78.8 | 85.9 | 471.3 | 73.8 | 94.7 | 97.7 | 59.6 | 87.9 | 94.5 | 508.2 |
| | BiCro* | 67.6 | 90.8 | 94.4 | 51.2 | 77.6 | 84.7 | 466.3 | 73.9 | 94.4 | 97.8 | 58.3 | 87.2 | 93.9 | 505.5 |
| | MSCN* | 70.4 | 91.0 | 94.9 | 53.4 | 77.8 | 84.1 | 471.6 | 74.4 | 95.1 | 97.9 | 59.2 | 87.1 | 92.8 | 506.5 |
| | **CRCL*** | **73.1** | **93.4** | **95.8** | **54.8** | **81.9** | **88.3** | **487.3** | **76.3** | **95.1** | **97.9** | **60.8** | **89.0** | **95.1** | **514.2** |
| 80% | SCAN | 5.1 | 18.1 | 27.3 | 3.9 | 13.1 | 19.1 | 86.6 | 22.2 | 51.9 | 67.5 | 13.8 | 41.1 | 58.6 | 255.1 |
| | SAF | 12.2 | 32.8 | 48.4 | 11.8 | 30.5 | 41.5 | 177.2 | 24.2 | 57.5 | 74.1 | 24.7 | 57.1 | 73.0 | 310.6 |
| | SGR | 13.7 | 35.1 | 47.6 | 12.1 | 30.9 | 41.9 | 181.3 | 26.7 | 60.7 | 75.6 | 25.3 | 58.2 | 72.6 | 319.1 |
| | VSE∞ | 8.1 | 23.1 | 34.7 | 7.4 | 22.6 | 31.8 | 127.7 | 25.4 | 55.1 | 70.6 | 19.2 | 50.5 | 68.0 | 288.8 |
| | NCR* | 1.4 | 7.1 | 11.7 | 1.5 | 5.4 | 9.3 | 36.4 | 21.6 | 52.6 | 67.6 | 15.1 | 38.1 | 49.8 | 244.8 |
| | DECL* | 60.7 | 84.6 | 91.2 | 42.1 | 69.6 | 78.6 | 426.8 | 65.6 | 91.6 | 96.6 | 52.0 | 83.0 | 91.3 | 480.1 |
| | BiCro* | 3.6 | 13.9 | 20.5 | 1.7 | 7.5 | 13.0 | 60.2 | 40.0 | 72.6 | 84.7 | 22.6 | 53.0 | 67.2 | 340.1 |
| | MSCN* | 1.0 | 4.4 | 9.1 | 0.4 | 1.4 | 2.5 | 18.8 | 66.8 | 91.6 | 96.2 | 52.7 | 83.0 | 90.9 | 481.2 |
| | **CRCL*** | **62.3** | **86.8** | **92.8** | **46.0** | **73.6** | **82.2** | **443.7** | **72.7** | **93.5** | **97.6** | **57.5** | **86.8** | **93.7** | **501.8** |

### 3.3.1 Results under Synthetic Noisy Correspondences

For quantitative evaluation under specific noise levels, we conduct all tested methods under four different noise rates (*i.e.*, 20%, 40%, 60%, and 80%) of synthetic noisy correspondences on the Flickr30K and MS-COCO datasets. Quantitative results on Flickr30K and MSCOCO 1K test set are shown in Table 1. For MS-COCO, the results are computed by a veraging over 5 folds of 1K test images. From the results, one can see that our CRCL could remarkably outperform the robust baselines (NCR, DECL, BiCro, and MSCN) on most of the metrics, which demonstrates the superior robustness of CRCL against NC. Moreover, our CRCL not only performs well in low noise but also achieves the best performance under high noise, especially 80% noise, which provides strong evidence for the stability and robustness of our method.

Table 2: Performance comparison on CC152K and MS-COCO 5K.

| | CC152K | | | | | | | MS-COCO 5K | | | | | | |
| | Image → Text | | | Text → Image | | | | Image → Text | | | Text → Image | | | |
| Methods | R@1 | R@5 | R@10 | R@1 | R@5 | R@10 | rSum | R@1 | R@5 | R@10 | R@1 | R@5 | R@10 | rSum |
|---|---|---|---|---|---|---|---|---|---|---|---|---|---|---|
| SCAN | 30.5 | 55.3 | 65.3 | 26.9 | 53.0 | 64.7 | 295.7 | 44.7 | 75.9 | 86.6 | 33.3 | 63.5 | 75.4 | 379.4 |
| VSE∞ | 34.0 | 64.5 | **77.0** | 12.9 | 19.2 | 21.6 | 229.2 | 56.6 | 83.6 | 91.4 | 39.3 | 69.9 | 81.1 | 421.9 |
| SGRAF | 32.5 | 59.5 | 70.0 | 32.5 | 60.7 | 68.7 | 323.9 | 58.8 | 84.8 | 92.1 | 41.6 | 70.9 | 81.5 | 429.7 |
| NCR* | 39.5 | 64.5 | 73.5 | 40.3 | 64.6 | 73.2 | 355.6 | 58.2 | 84.2 | 91.5 | 41.7 | 71.0 | 81.3 | 427.9 |
| DECL* | 39.0 | 66.1 | 75.5 | 40.7 | 66.3 | 76.7 | 364.3 | 59.2 | 84.5 | 91.5 | 41.7 | 70.6 | 81.1 | 428.6 |
| MSCN* | 40.1 | 65.7 | 76.6 | 40.6 | 67.4 | 76.3 | 366.7 | - | - | - | - | - | - | - |
| BiCro* | 40.8 | 67.2 | 76.1 | **42.1** | 67.6 | 76.4 | 370.2 | 59.0 | 84.4 | 91.7 | 42.4 | 71.2 | 81.7 | 430.4 |
| **CRCL*** | **41.8** | **67.4** | 76.5 | 41.6 | **68.0** | **78.4** | **373.7** | **61.3** | **85.8** | **92.7** | **43.5** | **72.6** | **82.7** | **438.6** |

### 3.3.2 Results under Real Noisy Correspondences

In addition to synthetic noise, we carry the comparison experiments on the real-world noisy dataset CC152K. The quantitative results are shown in Table 2. From the results, one can see that our CRCL is superior to all baselines with the best overall performance of 373.7%, which indicates that the proposed method is robust against real-world noise. Specifically, CRCL outperforms the best baseline BiCro, with absolute performance improvement of 1.0%, 0.2%, 0.4%, 0.4%, and 2.0% across different metrics, except for R@1 in text-to-image retrieval.

### 3.3.3 Results under Well-annotated Correspondences

Besides noisy correspondences, we also evaluate the tested methods trained on the well-aligned MS-COCO dataset for a comprehensive comparison. The results on MS-COCO 5K are shown in Table 2, wherein the results of all baselines are reported by the original papers for a fair comparison, except for the reproduced results of BiCro. From the table, our CRCL remarkably outperforms all baselines in terms of rSum. Specifically, CRCL prevails over the best baselines by 8.2% on overall performance (*i.e.*, rSum) absolutely, which shows that our CRCL is not only suitable for noisy cases but also performs well in well-aligned ones.

## 3.4 Comparison to pre-trained model

In this section, we compare our CRCL to the pre-trained model CLIP [38] to further evaluate its effectiveness in handling NC. CLIP is a well-known large pre-trained model that is trained from scratch on a dataset of 400 million image-text pairs collected from the Internet, which includes a large number of training pairs with real NCs. More specifically, following [12], we report the zero-shot results and fine-tuning results under 20% noise and compare them with that of CRCL-SGRAF in Table 3. From the results, CLIP shows a significant performance drop during fine-tuning under NC. On the contrary, the performance of our CRCL under 20% noise is even better than the zero-shot result of CLIP (ViT-L/14), which shows the strength and potential of our CRCL in dealing with NC.

Table 3: Comparison with CLIP on MS-COCO 5K.

| Noise Rate | Methods | Image → Text | | | Text → Image | | | |
| | | R@1 | R@5 | R@10 | R@1 | R@5 | R@10 | rSum |
|---|---|---|---|---|---|---|---|---|
| 0%, Zero-Shot | CLIP (ViT-L/14) | 58.4 | 81.5 | 88.1 | 37.8 | 62.4 | 72.2 | 400.4 |
| | CLIP (ViT-B/32) | 50.2 | 74.6 | 83.6 | 30.4 | 56.0 | 66.8 | 361.6 |
| 20%, Fine-tune | CLIP (ViT-L/14) | 36.1 | 61.3 | 72.5 | 22.6 | 43.2 | 53.7 | 289.4 |
| | CLIP (ViT-B/32) | 21.4 | 49.6 | 63.3 | 14.8 | 37.6 | 49.6 | 236.3 |
| | **Our CRCL** | **59.3** | **85.2** | **91.9** | **42.9** | **71.9** | **82.1** | **433.3** |

## 3.5 Ablation Study

In this section, we present ablation studies conducted on Flickr30K with 60% noise, as shown in Table 4. From the table, one can find that the full version of our CRCL achieves the best performance, which indicates that each component contributes to our method for performance improvement. By recasting $q$ with the rectified label $\hat{y}_{ii}$ using SCC, $\mathcal{L}_r$ shows higher potential against NC. Moreover,

through the combination of active learning and robust complementary learning, there is further performance improvement, which indicates the effectiveness of our ACL. Note that these results are obtained from a single model for a comprehensive evaluation, and are not ensemble results as shown in Tables 1 and 2.

Table 4: Ablation studies on Flickr30K with 60% noise.

| Configuration | | Image → Text | | | Text → Image | | | |
|---|---|---|---|---|---|---|---|---|
| loss | SCC | R@1 | R@5 | R@10 | R@1 | R@5 | R@10 | rSum |
| $\mathcal{L}_{acl}$ | ✓ | **70.5** | **91.3** | **95.6** | **52.5** | **79.4** | **86.8** | **476.1** |
| $\mathcal{L}_d$ | ✓ | 69.3 | 91.1 | 95.2 | 51.4 | 78.2 | 86.0 | 471.2 |
| $\mathcal{L}_r$ | ✓ | 67.7 | 91.2 | 95.0 | 52.3 | 79.0 | 86.4 | 471.6 |
| $\mathcal{L}_r(q=1)$ | | 25.2 | 53.9 | 65.6 | 20.3 | 45.0 | 57.2 | 267.2 |
| $\mathcal{L}_r(q=0)$ | | 63.5 | 89.7 | 93.8 | 48.2 | 74.5 | 82.0 | 451.7 |
| $\mathcal{L}_d$ | | 65.6 | 87.5 | 93.2 | 47.7 | 74.6 | 82.7 | 451.3 |

### 3.6 Visualization Analysis

To further investigate the generalization and robustness of CRCL, we visually study the retrieval performance of our CRCL-VSE∞, CRCL-SAF, and CRCL-SGR on Flickr30K and MS-COCO with varying noise rates in Figure 1(a,d). From the figure, one can see that CRCL can enhance the robustness of the original methods and perform well even under high noise. In addition, we provide visualization to intuitively showcase the effectiveness of the proposed SCC. Specifically, we visualize the distributions of both cross-modal similarities and corrected correspondences for noisy training pairs using CRCL-SGR on Flickr30K with 60% NCs. For comparison, we also visualize the results without SCC, where we replace $\hat{y}_{ii}$ with the prediction $\hat{p}^t(I_i, T_i)$ for the current iteration. The visualizations clearly indicate that SCC prevents the over-accumulation of errors in correspondence correction, thus verifying its effectiveness in mitigating the impact of NC.

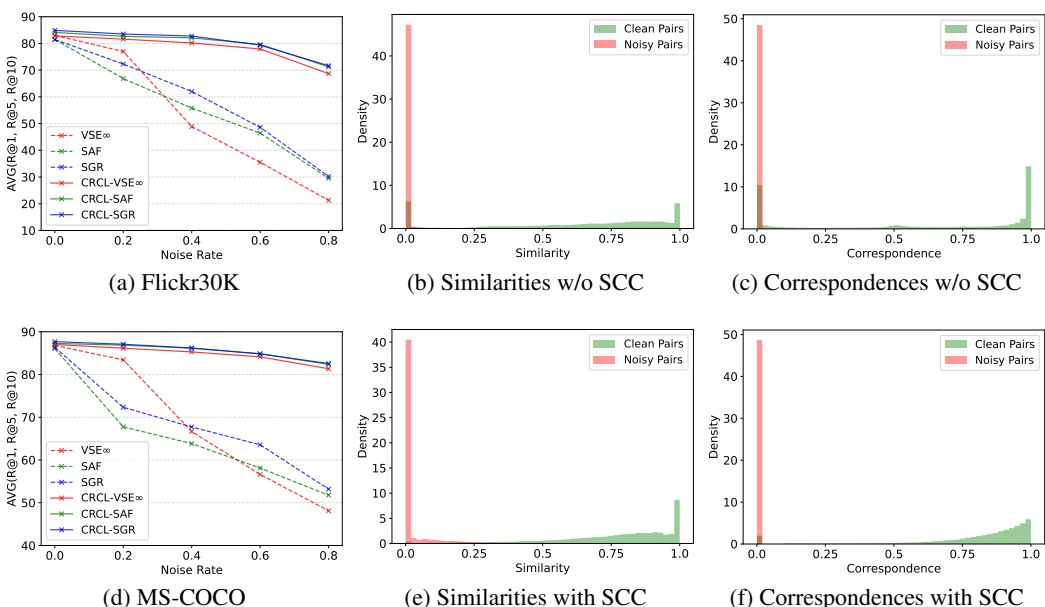

Figure 1: (a,d) The performance on Flickr30K and MS-COCO with varying noise rates; (b,c/e,f) The similarities and corrected correspondences of training pairs after learning without/with SCC.

## 4 Conclusion

In this paper, we propose a generalized robust framework CRCL to endow traditional methods with robustness against noisy correspondences. To alleviate the harmful effects brought by NC, we present

an Active Complementary Loss (ACL) and a Self-refining Correspondence Correction technique (SCC). These techniques enable effective learning of visual-semantic associations and progressive correction of correspondences based on historical predictions, thus boosting the robustness of image-text matching. Extensive experiments demonstrate that our CRCL achieves state-of-the-art robustness to both synthetic and real-world NCs. Furthermore, ablation studies and visualization analyses further verify the effectiveness and reliability of the proposed components.

## 5   Limitations and Broader Impact Statement

Despite the promising performance of our proposed CRCL, there are some limitations that should be acknowledged. First, we only study the NC problem between two modalities, *i.e.*, image and text. Second, we did not take into account category-level noise. The proposed CRCL likely impacts various applications that require robust image-text matching, *e.g.*, multimedia retrieval, and image annotation. We encourage further study to understand and mitigate the biases and risks potentially brought by image-text matching.

## Acknowledgments and Disclosure of Funding

This work was supported by the National Natural Science Foundation of China (U19A2078, U21B2040, 62372315, 62176171, and 62102274), Sichuan Science and Technology Planning Project (2023YFQ0020, 2023YFG0033, 2023ZHCG0016, 2022YFQ0014, 2022YFH0021), Chengdu Science and Technology Project (2023-XT00-00004-GX), the SCU-LuZhou Sciences and Technology Coorperation Program (2023CDLZ-16), and Fundamental Research Funds for the Central Universities under Grant YJ202140.

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
