# Supplementary Material

## for Cross-modal Active Complementary Learning with Self-refining Correspondence

**Yang Qin[1], Yuan Sun[1], Dezhong Peng[1,3,4], Joey Tianyi Zhou[2],
Xi Peng[1], Peng Hu[1]***

[1] College of Computer Science, Sichuan University, Chengdu, China.
[2] Centre for Frontier AI Research (CFAR) and Institute of High
Performance Computing (IHPC), A*STAR, Singapore.
[3] Chengdu Ruibei Yingte Information Technology Co., Ltd, Chengdu, China.
[4] Sichuan Zhiqian Technology Co., Ltd, Chengdu, China.
{qinyang.gm,joey.tianyi.zhou,pengx.gm,penghu.ml}@gmail.com,
sunyuan_work@163.com, pengdz@scu.edu.cn

In this supplementary material, we provide additional information for CRCL. Specifically, we first give detailed proofs of Equation (1) and Lemma 1 in Appendix A. To improve the reproducibility of CRCL, in Appendix B, we provide comprehensive implementation details of our CRCL for different extended baselines (*i.e.*, VSE∞[1], SAF[2], and SGR[2]) on three datasets. In addition, we present richer additional experimental results and analysis in Appendix C, including parameter analysis, progressive analysis, and extra comparison results, to fully verify the effectiveness and superiority of CRCL. Finally, we supplemented related work in Appendix D to further discuss the related research background.

## A Detailed Proofs

### A.1 Proof for Equation (1)

$$C \le R_{\mathcal{L}_r}(f^*) - R_{\mathcal{L}_r}(f^*_\eta) \le 0, \tag{1}$$

where $C = 2\eta(A_{\min}^{(1-q)} - A_{\max}^{(1-q)})/(1 - \frac{N\eta}{N-1}) \le 0$. $C$ increases as $q$ increases and when $q = 1$, $C$ takes the maximum value 0. $A_{\min}$ and $A_{\max}$ are the maximum and minimum values of $\sum_{j=1}^N \tan(p_{ij})$ under the condition $\sum_{j=1}^N p_{ij} = 1$, where $1 < A_{\min} < A_{\max}$, and $0 \le p_{ij} \le 1$ ($p_{ij} = p_{ij}^\circ$ or $p_{ij}^\diamond$). $f^*$ and $f^*_\eta$ are the global minimizers of $R_{\mathcal{L}_r}(f)$ and $R_{\mathcal{L}_r}^\eta(f)$, respectively.

*Proof.* Recall that for any $f$,

$$\begin{aligned}
R_{\mathcal{L}_r}(f) =& R_{\mathcal{L}_r^\circ}(f) + R_{\mathcal{L}_r^\diamond}(f) \\
=& \mathbb{E}_{(I_i,T_.)\sim\mathcal{D}}\left[y_{i\cdot}\mathcal{L}_r^\circ(I_i,T_.,q)\right] + \mathbb{E}_{(I_.,T_i)\sim\mathcal{D}}\left[y_{\cdot i}\mathcal{L}_r^\diamond(T_i,I_.,q)\right]. \\
=& \mathbb{E}_{(I_i,T_i)\sim\mathcal{D}}\left[\mathcal{L}_r^\circ(I_i,T_i,q)\right] + \mathbb{E}_{(I_i,T_i)\sim\mathcal{D}}\left[\mathcal{L}_r^\diamond(T_i,I_i,q)\right]
\end{aligned}$$

---

*Corresponding author.

37th Conference on Neural Information Processing Systems (NeurIPS 2023).

For uniform noisy correspondence with noise rate $\eta$, we consider the image-to-text direction and have

$$R^{\eta}_{\mathcal{L}^{\circ}_r}(f) = \mathbb{E}_{(I_i, T.) \sim \mathcal{D}_\eta} \left[ \tilde{y}_i \cdot \mathcal{L}^{\circ}_r(I_i, T., q) \right]$$

$$= \mathbb{E}_{(I_i, T.) \sim \mathcal{D}} \left[ (1 - \eta)\mathcal{L}^{\circ}_r(I_i, T_i, q) + \frac{\eta}{N - 1} \sum_{j \neq i} \mathcal{L}^{\circ}_r(I_i, T_j, q) \right]$$

$$= \mathbb{E}_{(I_i, T.) \sim \mathcal{D}} \left[ (1 - \eta)\mathcal{L}^{\circ}_r(I_i, T_i, q) + \frac{\eta}{N - 1} \left( (N - 1) \triangle^{(1-q)} - \mathcal{L}^{\circ}_r(I_i, T_i, q) \right) \right]$$

$$= \mathbb{E}_{(I_i, T.) \sim \mathcal{D}} \left[ (1 - \frac{N\eta}{N - 1})\mathcal{L}^{\circ}_r(I_i, T_i, q) + \eta\triangle^{(1-q)} \right],$$

where $\triangle = \sum_{j=1}^{N} \tan(p^{\circ}_{ij}) > 1$. Since $\triangle$ has the maximum and minimum values ($A_{\min}$ and $A_{\max}$, we provide a solution in **Remark**.) under the condition $\sum_{j=1}^{N} p^{\circ}_{ij} = 1, 0 \leq p^{\circ}_{ij} \leq 1$, for any $p^{\circ}_{ij}$, we have

$$(1 - \frac{N\eta}{N - 1})R_{\mathcal{L}^{\circ}_r}(f) + \eta A^{(1-q)}_{\min} \leq R^{\eta}_{\mathcal{L}^{\circ}_r}(f) \leq (1 - \frac{N\eta}{N - 1})R_{\mathcal{L}^{\circ}_r}(f) + \eta A^{(1-q)}_{\max}.$$

Similarly, the above equation also holds for $R_{\mathcal{L}^{\diamond}_r}(f)$ and $R^{\eta}_{\mathcal{L}^{\diamond}_r}(f)$, i.e.,

$$(1 - \frac{N\eta}{N - 1})R_{\mathcal{L}^{\diamond}_r}(f) + \eta A^{(1-q)}_{\min} \leq R^{\eta}_{\mathcal{L}^{\diamond}_r}(f) \leq (1 - \frac{N\eta}{N - 1})R_{\mathcal{L}^{\diamond}_r}(f) + \eta A^{(1-q)}_{\max}.$$

Thus, for $R^{\eta}_{\mathcal{L}_r}(f)$ and $R_{\mathcal{L}_r}(f)$, under $\eta \leq \frac{N-1}{N}$, we have

$$(1 - \frac{N\eta}{N - 1})R_{\mathcal{L}r}(f) + 2\eta A^{(1-q)}_{\min} \leq R^{\eta}_{\mathcal{L}r}(f) \leq (1 - \frac{N\eta}{N - 1})R_{\mathcal{L}r}(f) + 2\eta A^{(1-q)}_{\max}.$$

or equivalently,

$$(R^{\eta}_{\mathcal{L}r}(f) - 2\eta A^{(1-q)}_{\max})/(1 - \frac{N\eta}{N - 1}) \leq R_{\mathcal{L}r}(f) \leq (R^{\eta}_{\mathcal{L}r}(f) - 2\eta A^{(1-q)}_{\min})/(1 - \frac{N\eta}{N - 1}).$$

Thus, for $f^*_\eta$,

$$R_{\mathcal{L}_r}(f^*) - R_{\mathcal{L}_r}(f^*_\eta) \geq (R^{\eta}_{\mathcal{L}_r}(f^*) - R^{\eta}_{\mathcal{L}_r}(f^*_\eta))/(1 - \frac{N\eta}{N - 1}) + C \geq C, \tag{2}$$

or equivalently,

$$R^{\eta}_{\mathcal{L}_r}(f^*) - R^{\eta}_{\mathcal{L}_r}(f^*_\eta) \leq (1 - \frac{N\eta}{N - 1})(R_{\mathcal{L}_r}(f^*) - R_{\mathcal{L}_r}(f^*_\eta)) + C' \leq C', \tag{3}$$

where $C = 2\eta(A^{(1-q)}_{\min} - A^{(1-q)}_{\max})/(1 - \frac{N\eta}{N-1}) \leq 0$, $C' = 2\eta(A^{(1-q)}_{\max} - A^{(1-q)}_{\min}) \geq 0$, $f^*$ is a minimizer of $R_{\mathcal{L}_r}(f)$. Since $f^*_\eta$ and $f^*$ are the minimizers of $R^{\eta}_{\mathcal{L}_r}(f)$ and $R_{\mathcal{L}_r}(f)$, respectively, we have $R^{\eta}_{\mathcal{L}_r}(f^*) - R^{\eta}_{\mathcal{L}_r}(f^*_\eta) \geq 0$ or $R_{\mathcal{L}_r}(f^*) - R_{\mathcal{L}_r}(f^*_\eta) \leq 0$. Besides, it can be seen from Figure 1 that $C/C'$ increases/decreases as $q$ increases. In other words, under $\eta \leq \frac{N-1}{N}$, the larger $q$ is, the tighter the bound of Equation (3)/Equation (2) is. When $q$ is 1, then $R_{\mathcal{L}_r}(f^*) = R_{\mathcal{L}_r}(f^*_\eta)$ or $R^{\eta}_{\mathcal{L}_r}(f^*) = R^{\eta}_{\mathcal{L}_r}(f^*_\eta)$. This completes the proof.

**Remark** For the maximum value of $\sum_{j=1}^{N} \tan(p_{ij})$, For brevity, let $y = \sum_{j=1}^{N} \tan(p_{ij}) = \sum_{j=1}^{N} \tan(x_j)$ under the condition $\sum_{j=1}^{N} x_j = 1$ and $0 \leq x_j \leq 1$. For any $x_i, x_j \in [0, 1]$, we have

$$\tan(x_i + x_j) = \frac{\tan(x_i) + \tan(x_j)}{1 - \tan(x_i)\tan(x_j)}.$$

Since $0 \leq 1 - \tan(x_i)\tan(x_j) \leq 1$, we have

$$\tan(x_i + x_j) \geq (1 - \tan(x_i)\tan(x_j))\tan(x_i + x_j) = \tan(x_i) + \tan(x_j).$$

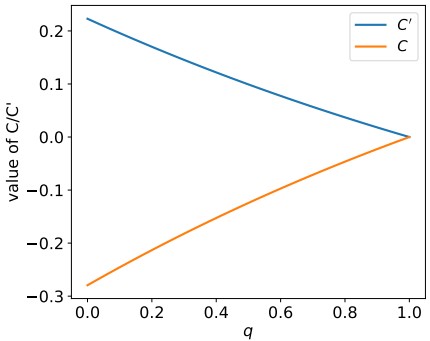

Figure 1: The value of $C/C'$ changes with $q$, wherein $N$ is 100 and $\eta$ is 0.2.

Hence, $\sum_{j=1}^{N} \tan(x_j) \leq \tan(\sum_{j=1}^{N} x_j) = \tan 1$, *i.e.*, $A_{max} = y_{max} = \tan 1 \approx 1.5574$.

For the minimum value of $y = \sum_{j=1}^{N} \tan(x_j)$, when $x_j \in (0,1)$, $\frac{\partial y}{\partial x_j} = \frac{1}{\cos^2 x_j} > 0$. Thus, the minimum value does not appear on the hyperplane boundary (0 or 1). We use the Lagrange Multiplier method [3] to construct the objective as follows

$$\min y, \ s.t. \sum_{j=1}^{N} x_j = 1 \Leftrightarrow \min_{x_j, \lambda} (y - \lambda(\sum_{j=1}^{N} x_j - 1)).$$

Let $f(x, \lambda) = y - \lambda(\sum_{j=1}^{N} x_j - 1) = \sum_{j=1}^{N} \tan(x_j) - \lambda(\sum_{j=1}^{N} x_j - 1)$, for $x_j, \lambda$, we have

$$\begin{cases} \frac{\partial f}{\partial x_j} = \frac{1}{\cos^2 x_j} - \lambda, & j = 1, 2, \cdots, N, \\ \frac{\partial f}{\partial \lambda} = \sum_{j=1}^{N} x_j - 1. \end{cases}$$

Let $\frac{\partial f}{\partial x_j} = \frac{\partial f}{\partial \lambda} = 0$, we have

$$\begin{cases} \lambda = \frac{1}{\cos^2 x_j}, & j = 1, 2, \cdots, N, \\ \sum_{j=1}^{N} x_j = 1. \end{cases}$$

Thus, when $x_1 = x_2 = \cdots = x_N = \frac{1}{N}$, $y$ has a minimum value, *i.e.*, $A_{min} = y_{min} = N \tan \frac{1}{N} > 1$.

$\square$

## A.2  Proof for Lemma 1

**Lemma 1.** *In an instance-level cross-modal matching problem, under uniform NC with noise rate $\eta \leq \frac{N-1}{N}$, when $q = 1$, $\mathcal{L}_r$ is noise tolerant.*

*Proof.* Recall that for any $f$,

$$\begin{aligned} R_{\mathcal{L}_r}(f) =& R_{\mathcal{L}_r^\circ}(f) + R_{\mathcal{L}_r^\diamond}(f) \\ =& \mathbb{E}_{(I_i, T.) \sim \mathcal{D}} [y_{i.} \mathcal{L}_r^\circ(I_i, T., q)] + \mathbb{E}_{(I., T_i) \sim \mathcal{D}} [y_{.i} \mathcal{L}_r^\diamond(T_i, I., q)]. \\ =& \mathbb{E}_{(I_i, T_i) \sim \mathcal{D}} [\mathcal{L}_r^\circ(I_i, T_i, q)] + \mathbb{E}_{(I_i, T_i) \sim \mathcal{D}} [\mathcal{L}_r^\diamond(T_i, I_i, q)] \end{aligned}$$

Under uniform noisy correspondence with noise rate $\eta$ and $q = 1$, for any $f$, $R^\eta_{\mathcal{L}^\circ_r}(f)$ is written as

$$
\begin{aligned}
R^\eta_{\mathcal{L}^\circ_r}(f) =& \mathbb{E}_{(I_i,T.)\sim\mathcal{D}_\eta}\left[\tilde{y}_i.\mathcal{L}^\circ_r(I_i,T.,q=1)\right] \\
=& \mathbb{E}_{(I_i,T.)\sim\mathcal{D}}[(1-\eta)\mathcal{L}^\circ_r(I_i,T_i,q=1)+\frac{\eta}{N-1}\sum_{j\neq i}\mathcal{L}^\circ_r(I_i,T_j,q=1)] \\
=& \mathbb{E}_{(I_i,T.)\sim\mathcal{D}}\left[(1-\eta)\mathcal{L}^\circ_r(I_i,T_i,q=1)+\frac{\eta}{N-1}\left((N-2)+\frac{\tan(p^\circ_{ii})}{\sum_{k=1}^N \tan(p^\circ_{ik})}\right)\right]- \quad (4) \\
=& \mathbb{E}_{(I_i,T.)\sim\mathcal{D}}\left[(1-\eta)\mathcal{L}^\circ_r(I_i,T_i,q=1)+\frac{\eta}{N-1}\left((N-1)-\mathcal{L}^\circ_r(I_i,T_i,q=1)\right)\right] \\
=& (1-\frac{N\eta}{N-1})R_{\mathcal{L}^\circ_r}(f)+\eta
\end{aligned}
$$

Note that the equation between $R^\eta_{\mathcal{L}^\diamond_r}(f)$ and $R_{\mathcal{L}^\diamond_r}(f)$ can also be derived similarly as Equation (4), *i.e.*, $R^\eta_{\mathcal{L}^\diamond_r} = (1 - \frac{N\eta}{N-1})R_{\mathcal{L}^\diamond_r}(f)+\eta$. Thus,

$$
R^\eta_{\mathcal{L}_r}(f) = (1 - \frac{N\eta}{N-1})R_{\mathcal{L}_r}(f) + 2\eta
$$

Now, for any $f$, $R^\eta_{\mathcal{L}_r}(f^*) - R^\eta_{\mathcal{L}_r}(f) = (1 - \frac{N\eta}{N-1})(R_{\mathcal{L}_r}(f^*) - R_{\mathcal{L}_r}(f)) \leq 0$, where $\eta \leq \frac{N-1}{N}$ and $f^*$ is a globalminimizer of $R_{\mathcal{L}_r}(f)$. This proves $f^*$ is also the global minimizer of $R^\eta_{\mathcal{L}_r}(f)$. $\qquad\square$

## B  Implementation Details

### B.1  Model Settings

In this section, we mainly detail the model settings and the implementation of CRCL. To comprehensively verify the effectiveness of our framework, we apply our CRCL to VSE$\infty$ [1], SAF [2], and SGR [2] for further robustness against NC, *i.e.*, CRCL-VSE$\infty$, CRCL-SAF, and CRCL-SGR. For the VSE model used in our CRCL-VSE$\infty$, we use the same encoder models as VSE$\infty$ [1] to project the local region features and word embeddings into the shared common space and then utilize GPO [1] to aggregate local representations into global representations, wherein the dimensionality of the common space is 1024. For the CRCL-SAF/SGR, like DECL [4], we directly perform our CRCL on the similarity output of these models without any changes to their models. In all experiments, we use the same image region features and text backbone for fairness. More specifically, we utilize a Faster R-CNN detection model [5] to extract local-level BUTD features of salient regions with top-36 confidence scores for each image, like [6, 2]. These features are encoded into a 2,048-dimensional feature vector and then projected into 1,024-dimensional image representations in the common space. For each text, the Bi-GRU language backbone encodes the word tokens into the same dimensional semantic vector space as the image representation. Following [1], we employ the size augmentation on the training data, which is then fed into the model. For all parameter settings, see Appendix B.2. The code of our CRCL will be released on GitHub.

### B.2  Parameter Settings

In this section, we fully provide the parameter settings of our experiments in Table 1 for easy reproducibility on three benchmark datasets, *i.e.*, Flickr30K, MS-COCO, and CC152K. We divide the parameter settings into two groups, the first group includes the parameter settings for the training without synthetic noise (=0%). The second group consists of the parameter settings for the training under synthetic noise ($> 0\%$). Simultaneously, each group details the training parameters of the three extensions of the baselines, *i.e.*, CRCL-VSE$\infty$, CRCL-SAF, and CRCL-SGR. Note that the result of CRCL-SGRAF in the paper is the ensemble results of CRCL-SAF and CRCL-SGR. Following [2, 4, 7], the ensemble strategy is averaging the similarities computed by the two models and then performing image-text matching. Next, we will describe these main parameters. $e_f$ represents the number of epochs to freeze the correspondence label, avoiding insufficient model training in the early stage from affecting the correction quality. $e_i$ in $[e_1, \cdots, e_m]$ is the number of training epochs for the $i$-th SR piece. During the last SR piece, CRCL decays the learning rate (lr_rate) by 0.1 in lr_update

epochs. $\tau$ and $\lambda$ are the temperature parameter and the scale factor in ACL loss, respectively. $\beta$ and $\epsilon$ are the momentum coefficient and the similarity threshold in SCC, respectively. For the parametric analysis of some hyper-parameters, see Appendix C.2 for more details.

Table 1: The settings of some key parameters for training on three datasets.

| Noise | Datasets | Methods | $e_f$ | $[e_1, \cdots, e_m]$ | lr_update | lr_rate | $\beta$ | $\tau$ | $\lambda$ | $\epsilon$ |
|---|---|---|---|---|---|---|---|---|---|---|
| Synthetic noise $= 0\%$ | CC152K | CRCL-VSE∞ | 2 | [7,7,7,32] | 17 | 0.0005 | 0.8 | 0.05 | 5 | 0.1 |
| | | CRCL-SAF | 2 | [7,7,7,42] | 20 | 0.0005 | 0.8 | 0.05 | 5 | 0.1 |
| | | CRCL-SGR | 2 | [7,7,7,42] | 20 | 0.0005 | 0.8 | 0.05 | 5 | 0.1 |
| | Flickr30K | CRCL-VSE∞ | 2 | [7,7,7,32] | 15 | 0.0005 | 0.8 | 0.05 | 5 | 0.1 |
| | | CRCL-SAF | 2 | [7,7,7,32] | 15 | 0.0005 | 0.8 | 0.05 | 5 | 0.1 |
| | | CRCL-SGR | 2 | [7,7,7,32] | 15 | 0.0005 | 0.8 | 0.05 | 5 | 0.1 |
| | MS-COCO | CRCL-VSE∞ | 2 | [4,4,4,22] | 12 | 0.0005 | 0.8 | 0.05 | 5 | 0.1 |
| | | CRCL-SAF | 2 | [4,4,4,22] | 12 | 0.0005 | 0.8 | 0.05 | 5 | 0.1 |
| | | CRCL-SGR | 2 | [4,4,4,22] | 12 | 0.0005 | 0.8 | 0.05 | 5 | 0.1 |
| Synthetic noise $> 0\%$ | Flickr30K | CRCL-VSE∞ | 2 | [7,7,7,32] | 15 | 0.0005 | 0.8 | 0.05 | 5 | 0.1 |
| | | CRCL-SAF | 2 | [7,7,7,32] | 15 | 0.0005 | 0.8 | 0.05 | 5 | 0.1 |
| | | CRCL-SGR | 2 | [7,7,7,32] | 15 | 0.0005 | 0.8 | 0.05 | 5 | 0.1 |
| | MS-COCO | CRCL-VSE∞ | 2 | [4,4,4,22] | 12 | 0.0005 | 0.8 | 0.05 | 5 | 0.1 |
| | | CRCL-SAF | 2 | [4,4,4,22] | 12 | 0.0005 | 0.8 | 0.05 | 5 | 0.1 |
| | | CRCL-SGR | 2 | [4,4,4,22] | 12 | 0.0005 | 0.8 | 0.05 | 5 | 0.1 |

# C  Additional Experiments and Analysis

## C.1  Parametric Analysis

The proposed CRCL has three sensitive key hyper-parameters, *i.e.*, the temperature parameter $\tau$, the momentum coefficient $\beta$, and the similarity threshold $\epsilon$. Thus, we conduct detailed parameter experiments (shown in Figure 2) on the Flickr30K dataset to evaluate the impact of different hyper-parameter settings and obtain better parameter settings for CRCL. Note that all parametric experiments are performed by CRCL-VSE∞ under 60% noise. As can be seen from Figure 2a, too large or too small $\tau$ both cause a performance drop. Thus, in all experiments, we recommend the range of $\tau$ is $0.03 \sim 0.07$. From Figure 2b, when the value of $\beta$ is set to two extreme values, *i.e.*, 0 and 1, the performance drops remarkably. Moreover, in the range of $(0, 1)$, as $\beta$ increases, the performance gradually improves. We think that with the increase of $\beta$, each correction performed by MC will retain more historical information to reduce perturbation. Thus providing more stable corrected correspondences for training. In all our experiments, $\beta$ is 0.8. From Figure 2c, we can see that proper filtering is beneficial for mitigating NC. We think this filtering strategy can prevent the active loss from exploiting these confident noisy pairs to produce more misleading gradients. Thus, we set $\epsilon$ as 0.1 in all our experiments.

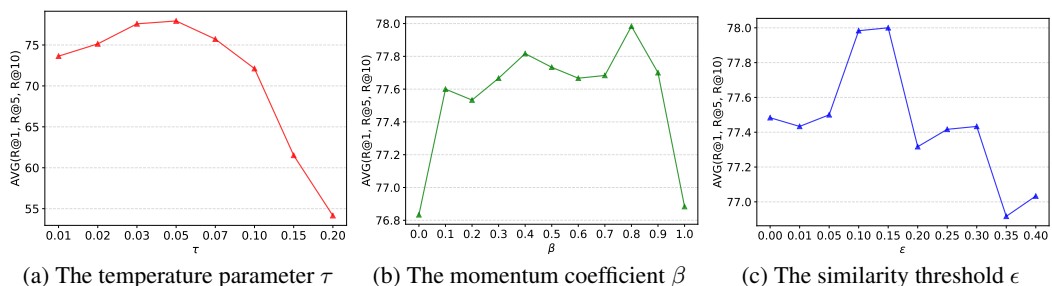

(a) The temperature parameter $\tau$    (b) The momentum coefficient $\beta$    (c) The similarity threshold $\epsilon$

Figure 2: Parametric analysis on Flickr30K with 60% noise.

## C.2  Progressive Analysis

To comprehensively investigate the effectiveness of our CRCL, we carry out some progressive processes to further analyze the advantages of CRCL. Specifically, we recorded the performance

of VSE$\infty$ with different loss functions, including CRCL-VSE$\infty$, $\mathcal{L}_d$, $\mathcal{L}_r(q = 1)$, $\mathcal{L}_r(q = 0)$, Complementary Contrastive Loss (CCL) [8], the hinge-based Triplet Ranking loss (TR) [9], the Triplet Tanking loss with Hard Negatives (TR-HN) [10], on Flickr30K under 80% noise. We visualize the performance of bidirectional retrieval in Figure 3. From the results, although $\mathcal{L}_r(q = 1)$ is noise-tolerant, which is consistent with the theoretical analysis (lemma 1), there would be some underfitting. Our CRCL with ACL loss fully explores the advantages of $\mathcal{L}_r$ and $\mathcal{L}_q$, showing remarkable robustness.

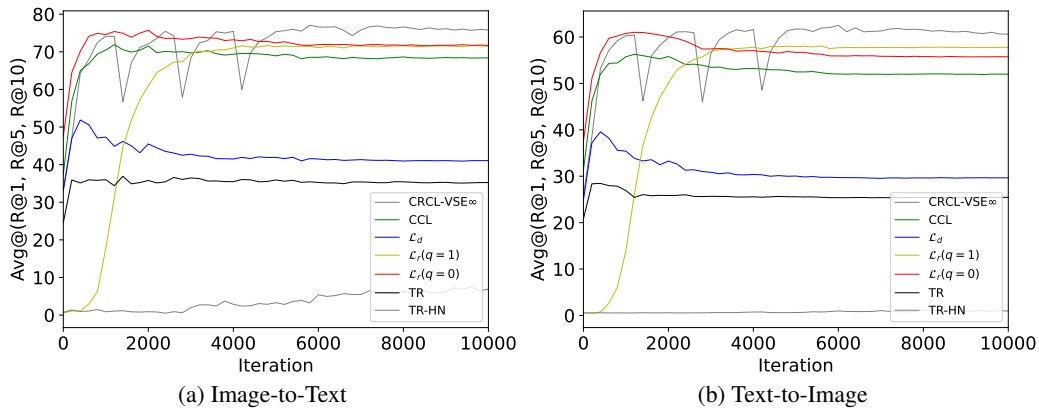

(a) Image-to-Text  (b) Text-to-Image

Figure 3: The performance of VSE$\infty$ with different loss functions.

### C.3 More Results under Synthetic Noisy Correspondences

To fully demonstrate the superiority and generalization of the proposed CRCL, we provide more comparison results under different robustness frameworks, including DECL[2] [4] and BiCro[3] [8]. In Table 2, except for the results of BiCro under 20%, 40%, and 60%, all other results are reproduced by us. From the results, our CRCL can significantly improve the robustness of existing methods (*e.g.*, VSE$\infty$, SAF, and SGR) and outperform other advanced robust frameworks. It is worth noting that CRCL is also stable and superior in high noise, which shows the effectiveness of our CRCL.

### C.4 More Results under Well-annotated Correspondences

In this section, we supplement the experimental results under well-annotated correspondences for a comprehensive and faithful comparison, including 17 state-of-the-art baselines, namely VSRN (ICCV'19) [11], CVSE (ECCV'20) [12], VSE$\infty$ (CVPR'21) [1], MV-VSE (IJCAI'22) [13]; SCAN (ECCV'18) [6], CAMP (ICCV'19) [14], IMRAM (CVPR'20) [15], GSMN (CVPR'20) [16], SGRAF (AAAI'21) [2], NCR (NeurIPS'21) [17], DECL (ACM MM'22) [4], CGMN (TOMM'22) [18], URDA (TMM'22) [19], CMCAN (AAAI'22) [20], NAAF (CVPR'22) [21], CCR&CCS (WACV'23) [22], RCL (TPAMI'23) [8], and BiCro (CVPR'23) [7]. From the experimental results in Table 3, our CRCL achieves competitive results, which demonstrates the ability and potential of CRCL to handle well-correspondence scenarios.

## D  Related works

### D.1  Image-Text Matching

Image-text matching methods mainly focus on learning latent visual-semantic relevance/similarities as the evidence for cross-modal retrieval [10, 6, 2, 1, 20, 23, 24, 25, 26, 27]. These approaches could be roughly classified into global- and local-level methods. To be specific, most global-level methods [10, 11, 1, 27] project images and texts into a shared global space, wherein cross-modal similarities could be computed [10, 1]. For example, Faghri et al. [10] proposed a triplet ranking

---

[2]https://github.com/QinYang79/DECL
[3]https://github.com/xu5zhao/BiCro

Table 2: Performance comparison (R@K(%) and rSum) of image-text retrieval on Flickr30K and MS-COCO 1K. The highest scores are shown in **bold**.

| Noise | Methods | Flickr30K | | | | | | | MS-COCO 1K | | | | | | |
| | | Image → Text | | | Text → Image | | | | Image → Text | | | Text → Image | | | |
| | | R@1 | R@5 | R@10 | R@1 | R@5 | R@10 | rSum | R@1 | R@5 | R@10 | R@1 | R@5 | R@10 | rSum |
| 20% | SAF | 51.8 | 79.5 | 88.3 | 38.1 | 66.8 | 76.6 | 401.1 | 41.0 | 78.4 | 89.4 | 38.2 | 74.0 | 85.5 | 406.5 |
| | SGR | 61.2 | 84.3 | 91.5 | 44.5 | 72.1 | 80.2 | 433.8 | 49.1 | 83.8 | 92.7 | 42.5 | 77.7 | 88.2 | 434.0 |
| | VSE∞ | 69.0 | 89.2 | 94.8 | 48.8 | 76.3 | 83.8 | 461.9 | 73.5 | 93.3 | 97.0 | 57.4 | 86.5 | 92.8 | 500.5 |
| | DECL-SAF | 73.1 | 93.0 | 96.2 | 57.0 | 82.0 | 88.4 | 489.7 | 77.2 | **95.9** | 98.4 | 61.6 | 89.0 | 95.3 | 517.4 |
| | DECL-SGR | 75.4 | 93.2 | 96.2 | 56.8 | 81.7 | 88.4 | 491.7 | 76.9 | 95.3 | 98.2 | 61.3 | 89.0 | 95.1 | 515.8 |
| | BiCro-SAF | **77.0** | 93.3 | 97.5 | 57.2 | 82.3 | 89.1 | 496.4 | 74.5 | 95.0 | 98.2 | 60.7 | 89.0 | 95.0 | 512.4 |
| | BiCro-SGR | 76.5 | 93.1 | 97.4 | 58.1 | 82.4 | 88.5 | 496.0 | 75.7 | 95.1 | 98.1 | 60.5 | 88.6 | 94.7 | 512.7 |
| | **CRCL-VSE∞** | 74.8 | 92.8 | 96.5 | 55.1 | 81.8 | 88.7 | 489.7 | 76.2 | 95.5 | **98.6** | 61.3 | 89.7 | 95.6 | 516.9 |
| | **CRCL-SAF** | 74.7 | 93.7 | 97.7 | 57.9 | 82.8 | 89.2 | 496.0 | 78.5 | 95.7 | 98.5 | 63.1 | 89.9 | 95.5 | 521.2 |
| | **CRCL-SGR** | 75.8 | **94.6** | **97.6** | **59.1** | **84.0** | **90.1** | **501.2** | **78.9** | 95.7 | 98.3 | **63.6** | **90.3** | **95.7** | **522.5** |
| 40% | SAF | 34.3 | 65.6 | 78.4 | 30.1 | 58.0 | 68.5 | 334.9 | 36.0 | 74.4 | 87.0 | 33.7 | 69.4 | 82.5 | 383.0 |
| | SGR | 47.2 | 76.4 | 83.2 | 34.5 | 60.3 | 70.5 | 372.1 | 43.9 | 78.3 | 89.3 | 37.0 | 72.8 | 85.1 | 406.4 |
| | VSE∞ | 30.2 | 58.3 | 70.2 | 22.3 | 49.6 | 62.7 | 293.3 | 53.3 | 84.3 | 92.1 | 31.4 | 63.8 | 75.0 | 399.9 |
| | DECL-SAF | 72.2 | 91.4 | 95.6 | 54.0 | 79.4 | 86.4 | 479.0 | 75.8 | 95.0 | 98.1 | 60.3 | 88.7 | 94.9 | 512.8 |
| | DECL-SGR | 72.4 | 92.2 | 96.5 | 54.5 | 80.1 | 87.1 | 482.8 | 75.9 | 95.3 | 98.2 | 60.2 | 88.3 | 94.8 | 512.7 |
| | BiCro-SAF | 72.5 | 91.7 | 95.3 | 53.6 | 79.0 | 86.4 | 478.5 | 75.2 | 95.0 | 97.9 | 59.4 | 87.9 | 94.3 | 509.7 |
| | BiCro-SGR | 72.8 | 91.5 | 94.6 | 54.7 | 79.0 | 86.3 | 478.9 | 74.6 | 94.8 | 97.7 | 59.4 | 87.5 | 94.0 | 508.0 |
| | **CRCL-VSE∞** | 71.2 | 92.6 | 96.3 | 53.2 | 80.4 | 87.4 | 481.1 | 74.4 | 95.1 | **98.4** | 59.5 | 89.1 | 95.2 | 511.7 |
| | **CRCL-SAF** | 74.2 | 93.8 | 97.1 | 57.0 | 81.8 | 88.6 | 492.5 | 76.4 | **95.7** | 98.1 | **62.1** | 89.3 | 95.3 | 516.9 |
| | **CRCL-SGR** | **75.5** | **94.0** | **97.8** | 57.5 | 82.6 | 89.2 | 496.6 | 76.8 | 95.3 | 98.2 | 61.9 | 89.6 | 95.4 | 517.2 |
| 60% | SAF | 28.3 | 54.5 | 67.5 | 22.1 | 47.3 | 59.0 | 278.7 | 28.2 | 63.9 | 79.4 | 31.1 | 65.6 | 80.5 | 348.7 |
| | SGR | 28.7 | 58.0 | 71.0 | 23.8 | 49.5 | 60.7 | 291.7 | 37.6 | 73.3 | 86.3 | 33.8 | 68.6 | 81.7 | 381.3 |
| | VSE∞ | 18.0 | 44.0 | 55.7 | 15.1 | 38.5 | 51.8 | 223.1 | 33.4 | 64.8 | 79.1 | 26.0 | 60.1 | 76.3 | 339.7 |
| | DECL-SAF | 66.4 | 88.1 | 93.6 | 49.8 | 76.1 | 84.4 | 458.4 | 71.1 | 93.6 | 97.3 | 57.9 | 86.8 | 93.8 | 500.5 |
| | DECL-SGR | 68.5 | 89.9 | 94.8 | 50.3 | 76.7 | 84.1 | 464.3 | 73.2 | 94.4 | 97.9 | 58.2 | 86.8 | 93.9 | 504.4 |
| | BiCro-SAF | 67.1 | 88.3 | 93.8 | 48.8 | 75.2 | 83.8 | 457.0 | 72.5 | 94.3 | 97.9 | 57.7 | 86.9 | 93.8 | 503.1 |
| | BiCro-SGR | 68.5 | 89.1 | 93.1 | 48.2 | 74.7 | 82.7 | 456.3 | 73.4 | 94.0 | 97.5 | 58.0 | 86.8 | 93.6 | 503.3 |
| | **CRCL-VSE∞** | 68.3 | 89.8 | **95.9** | 50.5 | 77.8 | 85.3 | 467.6 | 72.6 | 94.1 | **98.0** | 57.8 | 87.7 | 94.5 | 504.7 |
| | **CRCL-SAF** | 70.1 | 90.8 | 95.7 | **53.0** | 79.4 | **86.9** | 475.9 | 74.6 | 94.5 | 97.6 | **59.5** | **88.3** | 94.7 | **509.2** |
| | **CRCL-SGR** | **70.5** | **91.3** | 95.6 | 52.5 | **79.4** | 86.8 | **476.1** | 74.6 | **94.6** | 97.9 | 59.2 | 88.0 | 94.6 | 508.9 |
| 80% | SAF | 12.2 | 32.8 | 48.4 | 11.8 | 30.5 | 41.5 | 177.2 | 24.2 | 57.5 | 74.1 | 24.7 | 57.1 | 73.0 | 310.6 |
| | SGR | 13.7 | 35.1 | 47.6 | 12.1 | 30.9 | 41.9 | 181.3 | 26.7 | 60.7 | 75.6 | 25.3 | 58.2 | 72.6 | 319.1 |
| | VSE∞ | 8.1 | 23.1 | 34.7 | 7.4 | 22.6 | 31.8 | 127.7 | 25.4 | 55.1 | 70.6 | 19.2 | 50.5 | 68.0 | 288.8 |
| | DECL-SAF | 56.3 | 82.1 | 89.3 | 38.7 | 64.7 | 73.8 | 404.9 | 65.9 | 92.0 | 96.6 | 52.9 | 83.6 | 91.7 | 482.7 |
| | DECL-SGR | 55.1 | 79.8 | 87.2 | 37.4 | 63.4 | 72.9 | 395.8 | 65.6 | 91.6 | 96.6 | 52.0 | 83.0 | 91.3 | 480.1 |
| | BiCro-SAF | 2.4 | 9.1 | 15.8 | 2.4 | 8.3 | 13.7 | 51.7 | 39.6 | 72.6 | 84.7 | 22.4 | 52.8 | 67.1 | 368.9 |
| | BiCro-SGR | 1.7 | 8.7 | 13.7 | 1.3 | 5.1 | 8.9 | 39.4 | 31.4 | 62.0 | 75.2 | 30.0 | 60.7 | 73.2 | 332.5 |
| | **CRCL-VSE∞** | 55.3 | 82.1 | 89.1 | 39.7 | 68.2 | 77.8 | 412.2 | 67.9 | 92.8 | **97.1** | 53.1 | 84.7 | 92.5 | 488.1 |
| | **CRCL-SAF** | 58.4 | 83.9 | 90.5 | **44.1** | 70.7 | 79.8 | 427.4 | **70.9** | 92.8 | **97.1** | 55.2 | 85.3 | 92.9 | 494.2 |
| | **CRCL-SGR** | **59.2** | **85.1** | **91.1** | 43.6 | **70.9** | **80.1** | **430.0** | 70.7 | **92.9** | **97.1** | **56.0** | **85.6** | **93.1** | **495.4** |

loss with hard negatives to learn holistic visual-semantic embeddings for cross-modal retrieval. A Generalized Pooling Operator (GPO) [1] was proposed to adaptively aggregate different features (*e.g.*, region-based and grid-based ones) for better common representations. For the local-level methods, most of them desire to learn the latent fine-grained alignments across modalities for more accurate inference of visual-semantic relevance [6, 2, 20]. Representatively, Lee et al.[6] proposed a Stacked Cross Attention Network model (SCAN) to excavate the full latent alignments by contextualizing the image regions and word tokens for visual-semantic similarity inference. Diao et al. [2] proposed a Similarity Graph Reasoning and Attention Filtration model (SGRAF) for accurate cross-modal similarity inference by using a graph convolutional neural network for fine-grained alignments and an attention mechanism for representative alignments. Moreover, Zhang et al. [20] proposed a novel Cross-Modal Confidence-Aware Network to combine the confidence of matched region-word pairs with local semantic similarities for a more accurate visual-semantic relevance measurement. HREM [27] could explicitly capture both fragment-level relations within modality and instance-level relations across different modalities, leading to better retrieval performance. Pan et.al [26] propose a Cross-modal Hard Aligning Network (CHAN) to comprehensively exploit the most relevant region-word pairs and eliminate all other alignments, achieving better retrieval accuracy and efficiency. However, the aforementioned methods rely heavily on well-aligned image-text pairs while ignoring the inevitable noisy correspondences in data [17, 4], which will mislead the cross-modal learning and lead to performance corruption.

Table 3: Performance comparison (R@K(%) and rSum) of image-text retrieval on Flickr30K and MS-COCO 1K. The highest scores are shown in **bold**. * means global-level method.

| | Flickr30K | | | | | | | MS-COCO 1K | | | | | | |
| | Image → Text | | | Text → Image | | | | Image → Text | | | Text → Image | | | |
| Methods | R@1 | R@5 | R@10 | R@1 | R@5 | R@10 | rSum | R@1 | R@5 | R@10 | R@1 | R@5 | R@10 | rSum |
|---|---|---|---|---|---|---|---|---|---|---|---|---|---|---|
| VSRN* | 71.3 | 90.6 | 96.0 | 54.7 | 81.8 | 88.2 | 482.6 | 76.2 | 94.8 | 98.2 | 62.8 | 89.7 | 95.1 | 516.8 |
| CVSE* | 70.5 | 88.0 | 92.7 | 54.7 | 82.2 | 88.6 | 476.7 | 69.2 | 93.3 | 97.5 | 55.7 | 86.9 | 93.8 | 496.4 |
| VSE∞* | 76.5 | 94.2 | 97.7 | 56.4 | 83.4 | 89.9 | 498.1 | 78.5 | 96.0 | 98.7 | 61.7 | 90.3 | 95.6 | 520.8 |
| MV-VSE* | 79.0 | 94.9 | 97.7 | 59.1 | 84.6 | 90.6 | 505.9 | 78.7 | 95.7 | 98.7 | 62.7 | 90.4 | 95.7 | 521.9 |
| SCAN | 67.4 | 90.3 | 95.8 | 48.6 | 77.7 | 85.2 | 465.0 | 72.7 | 94.8 | 98.4 | 58.8 | 88.4 | 94.8 | 507.9 |
| CAMP | 68.1 | 89.7 | 95.2 | 51.5 | 77.1 | 85.3 | 466.9 | 72.3 | 94.8 | 98.3 | 58.5 | 87.9 | 95.0 | 506.8 |
| IMRAM | 74.1 | 93.0 | 96.6 | 53.9 | 79.4 | 87.2 | 484.2 | 76.7 | 95.6 | 98.5 | 61.7 | 89.1 | 95.0 | 516.6 |
| GSMN | 76.4 | 94.3 | 97.3 | 57.4 | 82.3 | 89.0 | 496.7 | 78.4 | 96.4 | 98.6 | 63.3 | 90.1 | 95.7 | 522.5 |
| SGRAF | 77.8 | 94.1 | 97.4 | 58.5 | 83.0 | 88.8 | 499.6 | 79.6 | 96.2 | 98.5 | 63.2 | 90.7 | 96.1 | 524.3 |
| NCR | 77.3 | 94.0 | 97.5 | 59.6 | 84.4 | 89.9 | 502.7 | 78.7 | 95.8 | 98.5 | 63.3 | 90.4 | 95.8 | 522.5 |
| DECL | 79.8 | 94.9 | 97.4 | 59.5 | 83.9 | 89.5 | 505.0 | 79.1 | 96.3 | 98.7 | 63.3 | 90.1 | 95.6 | 523.1 |
| CGMN | 77.9 | 93.8 | 96.8 | 59.9 | 85.1 | 90.6 | 504.1 | 76.8 | 95.4 | 98.3 | 63.8 | 90.7 | 95.7 | 520.7 |
| UARDA | 77.8 | 95.0 | 97.6 | 57.8 | 82.9 | 89.2 | 500.3 | 77.8 | 95.0 | 97.6 | 57.8 | 82.9 | 89.2 | 500.3 |
| CMCAN | 79.5 | 95.6 | 97.6 | 60.9 | 84.3 | 89.9 | 507.8 | 78.6 | 96.5 | **98.9** | 63.9 | 90.7 | 96.2 | 524.8 |
| NAAF | 78.3 | 94.1 | 97.7 | 58.9 | 83.3 | 89.0 | 501.3 | 78.9 | 96.0 | 98.7 | 63.1 | **91.4** | **96.5** | 524.6 |
| CCR&CCS | 79.3 | 95.2 | 98.0 | 59.8 | 83.6 | 88.8 | 504.7 | 80.2 | **96.8** | 98.7 | 64.3 | 90.6 | 95.8 | 526.4 |
| RCL | 79.9 | **96.1** | 97.8 | 61.1 | 85.4 | 90.3 | 510.6 | 80.4 | 96.4 | 98.7 | 64.3 | 90.8 | 96.0 | 526.6 |
| BiCro | **80.7** | 94.3 | 97.6 | 59.8 | 83.8 | 89.7 | 505.9 | 78.3 | 95.8 | 98.5 | 62.7 | 90.0 | 95.7 | 521.0 |
| **CRCL** | 78.5 | 95.5 | **98.0** | 62.3 | 86.5 | 91.7 | **512.5** | 80.7 | 96.5 | 98.6 | **65.1** | 91.2 | 96.1 | **528.2** |

## D.2 Learning with Noisy Labels

Since the lack of well-annotated data in many real-world applications [28, 29, 30, 31, 32, 33, 34, 17], learning with incomplete/noisy supervision information is becoming more and more popular in recent years. In this section, we briefly review a few families of these methods against noisy labels: **1) Robust losses** aims to improve the robustness of loss functions to prevent models from overfitting on noisy labels [28, 35, 36, 37, 38, 30]. **2) Sample selection** [39, 40] mainly exploits the memorization effect of DNNs [41] to divide/select the corrupted samples from datasets, and then conduct different training strategies for clean and noisy data. **3) Correction Approaches**[42, 43, 44] attempt to correct the wrong supervision information (*e.g.*, labels or losses) for robust training through some ingenious mechanisms. Different from the aforementioned unimodal category-based methods, learning with noisy correspondence focuses on the noisy annotations existing across different modalities instead of classes [17, 4]. That is to say, noisy correspondences are instance-level noise instead of class-level noise, which is more challenging [17, 4]. To tackle this challenge, Huang et al. [17] first proposed a novel Noisy Correspondence Rectifier (NCR) to rectify the noisy correspondences with co-teaching. By introducing evidential deep learning into image-text matching, Qin et al. [4] proposed a general Deep Evidential Cross-modal Learning framework (DECL) to improve the robustness against noisy correspondences. Some recent works [8, 7] try to predict correspondence labels to recast the margin of triplet ranking loss [10] as a soft margin to further improve robustness like NCR, *e.g.*, cross-modal mete learning [8] and similarity-based consistency learning [7]. In addition to image-text matching, other fields are also troubled by NC, such as partially view-aligned clustering [45, 46, 47, 48], video-text retrieval [49], visible-infrared person re-identification [50]. In this paper, we mainly focus on the NC problem in image-text matching and try to address this from both robust loss function and correspondence correction.