# OpenReview forum: "Cross-modal Active Complementary Learning with Self-refining Correspondence"
_NeurIPS.cc/2023/Conference — NeurIPS 2023 poster_

### Official Review · Reviewer_DZ4B · 2023-07-02

**Soundness:** 3 good
**Presentation:** 3 good
**Contribution:** 4 excellent
**Rating:** 7
**Confidence:** 5

**Summary:**

This paper tackles a new challenge in image-text matching, namely, noisy correspondence, which refers to the mismatched image-text pairs that can mislead the model to learn incorrect cross-modal associations during training, resulting in a suboptimal cross-modal model that computes inaccurate similarities for retrieval. To address this challenge, the authors propose a general cross-modal robust complementary learning framework CRCL that can enhance the robustness and performance of existing image-text matching methods. The authors provide theoretical and empirical evidence that their method is effective and achieves state-of-the-art performance under the same settings compared with the existing robust baselines, such as NCR, DECL, and BiCro.

**Strengths:**

1. This paper compares with a comprehensive and fair set of baselines, including the state-of-the-art robust method BiCro (CVPR’23). Moreover, the paper also includes CCR&CCS (WACV’23) and RCL (TPAMI’23) in the appendix, which is commendable.
2. This paper provides appropriate theoretical analysis in the appendix, which is satisfactory. The authors cleverly adapt the robust theory of the noisy label problem and views NC as an instance-level category noise label problem. The experimental results and the theoretical analysis are consistent and supportive of each other.
3. This paper shows promising performance on both synthetic NC and real NC datasets. Especially under high noise levels, CRCL demonstrates a strong robustness against NC.
4. From Figure 1, it is evident that SCC is effective in alleviating the accumulation of noise errors and improving the accuracy of rectified correspondence labels.


**Weaknesses:**

1. There is a typo in lines 48-49: CSRL should be CRCL.
2. This paper verifies the generality of CRCL on three standard methods, i.e., VSE$\infty$, SGR, and SAF. How about the training cost of these methods, e.g., compared to the existing robust frameworks BiCro, NCR, and DECL?
3. The paper seems to lack a discussion of related works in the main text, although I found it in the appendix. I understand that it may be due to the space limitation. However, I think CRCL is a significant improvement over NCR (NeurIPS’2021) for the problem of noisy correspondence. Therefore, for the readers who are not familiar with NCR or noisy correspondence, it would be more helpful to discuss related work in the main text.
4. Is it possible to add Plausible-Match R-Precision (PMRP)[1] as an evaluation metric, since it may be more suitable for many-to-many matching evaluation? Although I believe that Recall can reflect most of the retrieval performance, PMRP may be more appealing.
5. I think SCC is a feasible and effective technique. What is the intuition behind it? Can the authors provide more explanation?

[1] Chun S, Oh S J, De Rezende R S, et al. Probabilistic embeddings for cross-modal retrieval[C]//Proceedings of the IEEE/CVF Conference on Computer Vision and Pattern Recognition. 2021: 8415-8424.


**Questions:**

See Weaknesses.

**Limitations:**

The authors discussed the limitations and Broader Impacts of CRCL in section 5.

---

> ### Author Rebuttal · Authors · 2023-08-09
>
> Thanks for your valuable comments and insightful suggestions. We will address your concerns and questions one by one as follows.
>
> **Q1. here is a typo in lines 48-49: CSRL should be CRCL.**
>
> Thank you for your detailed review. We will correct all typos in the next version.
>
> **Q2. This paper verifies the generality of CRCL on three standard methods, i.e., VSE, SGR, and SAF. How about the training cost of these methods, e.g., compared to the existing robust frameworks BiCro, NCR, and DECL?**
>
> Thanks for your valuable comment. We performed an efficiency analysis following DECL. For a fair comparison, all experiments are performed on a single GeForce RTX3090 24GB GPU and the same training environment. We report the results on Flickr30K with 20% NC as follows:
>
> |Method|Video memory (GB)|Time/epoch|rSum
> |-|-|-|-
> |NCR|11.9 GB|45.2 min|486.8
> |DECL-SAF|10.9 GB|13.5 min|479.0
> |DECL-SGR|14.6 GB|18.3 min|482.8
> |BiCro-SAF|8.1 GB|15.6 min|478.5
> |BiCro-SGR|11.8 GB|20.0 min|478.9
> |CRCL-VSE$\infty$|3.3 GB|1.3 min|481.1
> |CRCL-SAF|7.3 GB|12.9 min|492.5
> |CRCL-SGR|10.4 GB|15.2 min|496.6
>
> From the experimental results, one can see that our method is more efficient. Especially, compared with the co-teaching architecture NCR and BiCro, our CRCL is more efficient and has a better performance.
>
> **Q3. About the related works.**
>
> Thank you for your valuable suggestion. Due to space constraints, we have to move the related works to the supplementary material. If possible, in the next version, we will try to include the related works in the main text.
>
> **Q4. Is it possible to add Plausible-Match R-Precision (PMRP)[1] as an evaluation metric, since it may be more suitable for many-to-many matching evaluation? Although I believe that Recall can reflect most of the retrieval performance, PMRP may be more appealing.**
>
> Thank you for your valuable suggestions. We report the results with the PMRP metric on the MS-COCO dataset with 40% noise, as follows (**The first four columns are the results on the MS-COCO 5-fold 1K test, and the last four columns are the results on MS-COCO 5K test.**):
>
> ||I2T|I2T|T2I|T2I|I2T|I2T|T2I|T2I
> |-|-|-|-|-|-|-|-|-
> |Methods|R@1|PMRP|R@1|PMRP|R@1|PMRP|R@1|PMRP
> |PCME (0% noise) [1]|68.8|45.1|54.6|46.0|44.2|34.1|31.9|34.4
> |DECL-SAF|75.8|46.4|60.3|47.8|54.1|35.4|38.8|36.0
> |CRCL-SAF|76.4|46.4|62.1|47.9|54.8|35.5|40.2|36.1
> |DECL-SGR|75.9|46.0|60.2|46.7|54.3|30.8|38.5|35.0
> |CRCL-SGR|76.8|46.9|61.9|48.3|55.7|35.5|40.3|36.4
> |DECL-SGRAF|77.1|47.0|61.5|48.2|55.8|36.0|40.1|36.4
> |CRCL-SGRAF|78.2|47.3|63.3|48.7|57.6|36.0|41.7|36.8
>
> From these results, our CRCL not only achieves better Recall@1 accuracies but also has higher PMRP scores for bidirectional retrieval.
>
> **Q5. I think SCC is a feasible and effective technique. What is the intuition behind it? Can the authors provide more explanation?**
>
> Thank you for your insightful suggestion. In fact, our design intuition for SCC comes from addressing the flaws in previous works. Due to the memorization effect of DNNs, cross-modal learning will face the self-sample-selection error accumulation problem, which will degrade the performance. To alleviate the error accumulation, previous robust frameworks (e.g., NCR and BiCro) utilize the co-teaching manner to obtain accurate predictions.  However, the co-teaching strategy would increase the number of models, which will largely increase the training overhead. To address this problem, we present an efficient Self-refining Correspondence Correction (SCC) technique without any additional model to relieve the error accumulation problem. SCC leverages momentum correction to aggregate historical predictions of one cross-modal model, providing stable and accurate correspondence estimations while alleviating the over-memorization to NC. Moreover, SCC combines multiple independent self-refining processes in the entire training life, which could alleviate error accumulation against noisy correspondences.
>
> > [1] Probabilistic embeddings for cross-modal retrieval, CVPR'21.

---

> > ### Comment · Reviewer_DZ4B · 2023-08-19
> >
> > I have carefully read the authors’ rebuttal and the other reviewers’ comments. I think the authors have satisfactorily addressed all my concerns and improved the quality of their paper. Therefore, I maintain my positive score for this paper.

---

> > > ### Author Response · Authors · 2023-08-19
> > >
> > > We are very grateful for your feedback and positive recognition of our work. Your constructive comments have greatly enhanced the quality of our paper. We will revise our paper according to the reviews in the next version. Thank you again for your review and valuable time!

---

### Official Review · Reviewer_tALj · 2023-07-05

**Soundness:** 3 good
**Presentation:** 4 excellent
**Contribution:** 3 good
**Rating:** 6
**Confidence:** 5

**Summary:**

This paper tackles a latent challenge in image-text matching, which is the presence of noisy correspondences between images and texts. The paper introduces a general framework that combines a robust loss function and a correspondence correction technique to enhance the existing models’ ability to cope with noisy correspondences. The paper shows that the proposed CRCL framework outperforms the state-of-the-art methods. The paper also provides both experimental and theoretical evidence to support the effectiveness of the proposed algorithms.

**Strengths:**

Significance: Noisy correspondences are a common issue in image-text data, which can arise from alignment errors or weak cross-modal information. Studying how to deal with noisy correspondences can enable more applications of image-text learning, where the alignments are challenging and require domain expertise. This work makes significant contributions both theoretically and empirically, and offers a new perspective for future research on noisy correspondences and related fields.

Clarity: This paper is well-written and clear, and the motivation and method are easy to follow. The technical proofs seem to be sound. However, there are some typos, undefined terms, and missing references in the paper. Please see Weaknesses and Questions below.

Originality: The techniques used in this paper seem to be novel in the image-text matching field, and there are some technical innovations.

Quality: I think the quality of this paper is high, and I did not find any major flaws. The authors conduct comprehensive experiments to demonstrate the effectiveness of the proposed method.


**Weaknesses:**

- The related work section is incomplete. The authors should include and compare with some recent works on image-text matching, such as [1,2,3].
- Some typos: CRCL in line 48.
- Eq.1 is confusing. I suggest rewriting it as $1 - y_{ik}$, with probability $\bar{\eta}_{ik}, \forall k\neq j$.
- Eq.11 is not clear. The authors should explain the meaning and derivation of each term.


[1] Huang Y, Wang Y, Zeng Y, et al. MACK: multimodal aligned conceptual knowledge for unpaired image-text matching[J]. Advances in Neural Information Processing Systems, 2022, 35: 7892-7904.\
[2] Goel S, Bansal H, Bhatia S, et al. Cyclip: Cyclic contrastive language-image pretraining[J]. Advances in Neural Information Processing Systems, 2022, 35: 6704-6719.\
[3] Pan Z, Wu F, Zhang B. Fine-Grained Image-Text Matching by Cross-Modal Hard Aligning Network[C]//Proceedings of the IEEE/CVF Conference on Computer Vision and Pattern Recognition. 2023: 19275-19284.


**Questions:**

- The authors should explain why their results are different from the previous works NCR and DECL. Are there any special settings or hyperparameters that affect the performance? The authors should also report the training efficiency of their method, such as the training time and the computational resources, and compare with the existing methods, such as DECL.
- The authors should justify their choice of using MS-COCO and Flickr30K datasets, which have one-to-many correspondences (1 to 5), while their problem formulation assumes one-to-one correspondences. How does this affect the validity and applicability of their method?
- The authors claim that ACL is a general framework that can be applied to any existing model. However, they do not provide any empirical evidence to support this claim under other tasks. For example, can ACL handle noisy labels? The authors should provide more experimental support for the generality of ACL.
- For Eq.11, it is not clear how the initial labels are obtained. Is it based on the first epoch of the first SR? The authors should clarify this point and explain how they initialize the labels.
- What does active loss mean? Is it the same as normalized loss [1]? The authors should define this term clearly.


[1] Ma X, Huang H, Wang Y, et al. Normalized loss functions for deep learning with noisy labels[C]//International conference on machine learning. PMLR, 2020: 6543-6553.


**Limitations:**

The authors have discussed the potential limitations and implications of their work.

---

> ### Author Rebuttal · Authors · 2023-08-09
>
> Thanks for your valuable comments and insightful suggestions. Attached is our point-by-point response.
>
> **Q1. Missing related works [1,2,3].**
>
> We appreciate your valuable feedback and the related works you mentioned.  We will include a discussion and a comparison of these works in the next version. However, due to the time limitation, we are not able to conduct the experiments under our settings. The comparison experiments with [3] can be found in the response (Q.6) to Reviewer QLFm.
>
> **Q2. Eq.11 is not clear.**
>
> Eq.11 is a mathematical description of updating binary correspondence labels in multiple SR pieces. Specifically, the next SR uses the updated labels from the previous SR piece as the initial labels for training. Each SR does not update the label before the $e_f$-th epoch, and the rest uses Momentum Correction to update. The only exception is that the first SR uses the predicted matching probabilities at the $e_f$-th epoch as the labels for momentum update.
>
> **Q3. (a) The explanation of why the results are different from the previous works NCR and DECL. (b) The training efficiency comparison between CRCL and DECL.**
>
> We appreciate your valuable feedback and constructive suggestions. **(a)** This difference in results is mainly caused by the difference in noise settings, i.e., the generation method of noisy correspondences and the noise rates. To fairly evaluate our method, we use the same settings and reproduce some results, which may differ from the results in the original papers of baselines. Specifically, we use the generation method used in NCR [12], i.e., **randomly shuffling the captions of training images for a specific percentage**. For a unified comparison, we set four noise rates, i.e., 20%, 40%, 60%, and 80%. Some of the results in the paper come from original papers, e.g., the results of BiCro (0%, 20%, 40%, and 60%) and NCR (0%, 20%) on Flickr30K and MS-COCO, and the results on CC152K. The others are reproduced by us under the same setting. **(b)** Due to the space limitation, the reply to this question can be found in the response (Q.2) to Reviewer DZ4B.
>
> **Q4. The authors should justify their choice of using MS-COCO and Flickr30K datasets, which have one-to-many correspondences (1 to 5), while their problem formulation assumes one-to-one correspondences. How does this affect the validity and applicability of their method?**
>
> Thanks a lot for your valuable comment. Although there are one-to-many correspondences in MS-COCO and Flickr30K datasets, they do not affect the validity and applicability of our method. In practice, we can only train the models by randomly sampling small mini-batches due to the limitation of computing resources. Moreover, the input image-text pairs in each mini-batch basically have one-to-one correspondences due to the sparsity of random sampling, which actually does not violate the assumptions of our problem formulation. Besides, from the experimental results, our CRCL is applicable and effective on MS-COCO and Flickr30K datasets with state-of-the-art performance.
>
> **Q5. (a) Further explanation about ACL is a general framework. (b) Can ACL handle noisy labels? (c) The authors should provide more experimental support for the generality of ACL.**
>
> We appreciate your thoughtful comment. **(a)** In this paper, we mainly focus on image-text matching, and apply our ACL to three models to demonstrate its generality, i.e., SAF, SGR, and VSE$\infty$. In the rebuttal phase, we have further applied our ACL to a new task (i.e., noisy labels) to demonstrate its generality as shown in the following table. **(b)** Yes, ACL can handle noisy labels due to the similarity between noisy correspondence and noisy labels. Specifically, we can equate positive pairs to positive classes, such that binary correspondence labels and class labels are actually equivalent. **(c)** To verify this, we use the released code of [4] to conduct a preliminary experiment and compare the performance of ACL with some baselines. Specifically, to adapt to noisy labels, we directly replace the bidirectional matching probabilities used in ACL with unidirectional class probabilities output by the $softmax$ layer. Since ACL needs to have a corrected label $\hat{y}$ to recast $q$, we simply replace $\hat{y}$ with the positive class probabilities (possibly false positive class). The comparison baselines are CE, GCE, and NCE + MAE in [4]. The results ($mean±std$)  are reported on CIFAR-10 over 3 random runs, as shown in the following table:
>
> |Mthods|0.2|0.4|0.6|0.8
> |-|-|-|-|-
> |CE|75.90±0.28|60.28±0.27|40.90±0.35|19.65±0.46
> |GCE|87.27±0.21|83.33±0.39|72.00±0.37|29.08±0.80
> |NCE+MAE|87.12±0.21|84.19±0.43|77.61±0.05|49.62±0.72
> |Our ACL|88.64±0.11|85.31±0.14|79.06±0.55|54.26±0.89
>
> From the results, our ACL shows superior performance and robustness in dealing with noisy labels, which demonstrates that ACL has the generality to handle noisy labels.
>
> **Q6. For Eq.11, it is not clear how the initial labels are obtained.**
>
> Thank you for your careful review. To clarify, the first SR is trained with the initial labels set as 1 and uses the predicted matching probabilities at the $e_f$-th epoch as the labels for subsequent momentum update, as described in Eq.11.
>
> **Q7. What does active loss mean? Is it the same as normalized loss [4]? The authors should define this term clearly.**
>
> Thank you for your valuable review. Yes, the active loss is inspired by the normalized loss [4]. We migrated this class-level definition (i.e., “active”) to instance-level image-text matching, i.e., a loss is defined as “active” if it only optimizes at positive pairs.
>
> >[1] MACK: multimodal aligned conceptual knowledge for unpaired image-text matching, NeurIPS'22.\
> [2] Cyclip: Cyclic contrastive language-image pretraining. NeurIPS'22.\
> [3] Fine-Grained Image-Text Matching by Cross-Modal Hard Aligning Network, CVPR'23.\
> [4] Normalized loss functions for deep learning with noisy labels, ICML'20.

---

### Official Review · Reviewer_QLFm · 2023-07-05

**Soundness:** 4 excellent
**Presentation:** 3 good
**Contribution:** 3 good
**Rating:** 8
**Confidence:** 5

**Summary:**

This paper presents a novel framework (CRCL) for cross-modal correspondence learning that can handle noisy image-text pairs. The key idea of CRCL is to use a complementary active loss (ACL) that balances between discriminative learning and robust learning. ACL leverages the rectified correspondence labels to adjust the loss function and assign active loss weights, which means that the potentially noisy pairs are more focused on robust learning. To obtain accurate rectified correspondence labels, the paper proposes a self-refining correspondence correction technique (SCC) that estimates the correlation between modalities and avoids error accumulation. The proposed framework is simple yet effective in mitigating the negative impact of noisy training pairs and achieving robust image-text matching. The paper also conducts extensive experiments on three benchmark datasets to demonstrate the effectiveness and rationality of the proposed method.

**Strengths:**

1. The CRCL framework seems to be a general and flexible approach that can be easily integrated with existing methods to enhance the robustness of image-text matching.

2. This paper is well-written and easy to follow. The authors provide clear explanations and motivations for their method.

3. The authors simplify the training process by discarding the co-teaching scheme used in NCR and BiCro, and instead focus on the loss function and the correction technique to make the method concise and clear.

4. The experimental results are impressive and convincing. It is also noteworthy that CRCL can improve the performance of the pre-trained model, e.g., CLIP, as shown in Table 3.


**Weaknesses:**

1. In Eq.3, there is a typo: $i_j$ should be $I_j$.

2. In Eq.11, some symbols are not clearly defined, e.g., $\hat{p}^(j,t-1)(*)$. Please explain their meanings and notations.

3. The NC problem can be viewed as a special case of the PMP problem. RCL[20] uses complementary contrastive learning (CCL) to deal with PMPs. Similarly, CRCL can also be regarded as a further extension and study of CCL. However, the paper does not provide a direct comparison between CRCL and RCL. It would be interesting to see how they differ in performance and insights.

4. In the supplementary material, there is a capitalization error: “For brevity” should be “for brevity”.

5. Why not include MSCN[A] as a baseline? MSCN seems to be from the same period as BiCro (CVPR’23) and also addresses the NC problem.

6. Some related works are missing: [B-C]. These papers also propose methods for image-text matching and could be relevant for comparison or discussion.

[A] Noisy Correspondence Learning With Meta Similarity Correction, CVPR, 2023.
[B] Learning Semantic Relationship Among Instances for Image-Text Matching, CVPR, 2023.
[C] Fine-Grained Image-Text Matching by Cross-Modal Hard Aligning Network, CVPR, 2023.


**Questions:**

This paper is well-written and easy to follow. It has all the essential elements of a good paper and I do not have any major objections to accept it. I appreciate the authors’ approach, which is more concise and theoretically sound than the existing robust image-text matching methods. However, there are some minor issues that need to be addressed, such as some typos and some recent related works that need to be further discussed. Noisy correspondence is an important research direction in the field of multimodality, as it is similar to noisy label learning. I hope the authors can not only improve the performance of their method, but also explore the potential impact of CRCL on more correspondence learning tasks.

**Limitations:**

The authors should enrich further, as described in the Questions section.

---

> ### Author Rebuttal · Authors · 2023-08-09
>
> Thanks for your valuable comments and insightful suggestions. We will address your concerns and questions one by one as follows.
>
> **Q1. In Eq.3, there is a typo: $i_j$  should be $I_j$.**
>
> Thank you for your careful review. We will correct it in the next version.
>
> **Q2. In Eq.11, some symbols are not clearly defined, e.g., $\hat{p}^{(j,t-1)}(*)$. Please explain their meanings and notations.**
>
> Thank you for your suggestion. $\hat{p}^{(j,t-1)}(I_i,T_i)$ denotes the average matching probability $(p_{ii}^\circ + p_{ii}^\diamond)/2$ of pair $(I_i,T_i)$ at the $(t-1)$-th epoch during the $j$-th SR training piece. We will supplement it in the next version.
>
> **Q3. Comparison results with RCL.**
>
> Thank you for your valuable suggestion. We followed the same experimental setup in RCL [20] to evaluate our method and compare them. Due to the time limitation, we only report partial results of the SGRAF variants on the Flickr30K dataset, as follows:
>
> |||I2T|I2T|I2T|T2I|T2I|T2I|||
> |-|-|-|-|-|-|-|-|-|-
> |Noise|Methods|R@1|R@5|R@10|R@1|R@5|R@10|rSum|$\uparrow$(%)
> |40%|RCL-SAF|68.8|89.8|95.0|51.0|76.7|84.8|466.1|
> |40%|CRCL-SAF|71.9|91.9|95.3|54.1|79.6|86.8|479.6|+13.5%
> |40%|RCL-SGR|71.3|91.1|95.3|51.4|78.0|85.2|472.3|
> |40%|CRCL-SGR|73.7|93.0|96.2|53.9|80.5|88.1|485.4|+13.1%
> |80%|RCL-SAF|45.0|72.8|80.8|30.7|56.5|67.3|353.1|
> |80%|CRCL-SAF|49.7|76.6|86.0|34.8|62.5|72.8|382.4|+29.3%
> |80%|RCL-SGR|47.1|70.5|79.4|30.3|56.1|66.3|349.7|
> |80%|CRCL-SGR|49.3|77.4|85.9|34.0|62.8|72.3|381.7|+32.0%
>
> From the above results, CRCL outperforms RCL on all metrics, especially on 80% noise. This demonstrates the effectiveness of our method in handling noisy correspondences (NC). RCL uses complementary contrastive learning (CCL) to provide robustness against NC, but as we discussed in the *Introduction* section, it does not explicitly mitigate the effect of easily separable noise to further improve performance. In contrast, our CRCL leverages SCC to provide more reasonable learning objectives for higher performance.
>
>
> **Q4. In the supplementary material, there is a capitalization error: “For brevity” should be “for brevity”.**
>
> Thank you for your careful review. We will correct it in the next version.
>
> **Q5. Why not include MSCN[A] as a baseline? MSCN seems to be from the same period as BiCro (CVPR’23) and also addresses the NC problem.**
>
> We appreciate your valuable comments and helpful suggestions. We agree that MSCN is a strong baseline and should be compared with our CRCL. However, due to the different noise settings, we cannot directly use the reported results of MSCN and need to rerun it under our settings. Unfortunately, training MSCN is very time- and resource-consuming. For instance, training a model of MSCN on MSCOCO requires 4$\times$V100 32GB cards for 20 days. Therefore, we were not able to finish all the experiments before the submission deadline. After the deadline, we did our best to complete the experiments and report them in the following table. **In the table, the first four columns are the results on Flickr30K, and the last four columns are the results on MS-COCO 1K test.**
>
> |Noise|Methods|I2T(R@1)|$\uparrow $(%)|T2I(R@1)|$\uparrow $(%)|I2T(R@1)|$\uparrow $(%)|T2I(R@1)|$\uparrow $(%)
> |-|-|-|-|-|-|-|-|-|-
> |20%|MSCN|77.4||59.6||78.1||64.3|
> |20%|CRCL|77.9|+0.5%|60.9|+1.3%|79.6|+1.5%|64.7|+0.4%
> |40%|MSCN|74.4||57.2||74.8||60.3|
> |40%|CRCL|77.8|+4.4%|60.0|+2.8%|78.2|+3.4%|63.3|+3.0%
> |60%|MSCN|70.4||53.4||74.4||59.2||
> |60%|CRCL|73.1|+2.7%|54.8|+1.4%|76.3|+1.9%|60.8|+1.6%
> |80%|MSCN|1.0||0.4||66.8||52.7|
> |80%|CRCL|62.3|+61.3%|46.0|+45.6%|72.7|+5.9%|57.5|+4.8%
>
> As can be seen from the table, our CRCL consistently outperforms MSCN on both datasets and under all noise levels, which also demonstrates the effectiveness and robustness of our CRCL against NC. Please note that the above table only includes partial comparison results in terms of Recall@1 due to the space limitation, and more experimental results of MSCN will be provided in the next version.
>
> **Q6. Some related works are missing: [B-C]. These papers also propose methods for image-text matching and could be relevant for comparison or discussion.**
>
> We appreciate your feedback and the related works you mentioned. We will include a discussion of these works in the next version. Briefly, HREM [B] could explicitly capture both fragment-level relations within modality and instance-level relations across different modalities, leading to better retrieval performance. Pan et.al [C] propose a Cross-modal Hard Aligning Network (CHAN) to comprehensively exploit the most relevant region-word pairs and eliminate all other alignments, achieving better retrieval accuracy and efficiency. However, unlike them, our CRCL is a specific framework to address NC. Despite this, compared with them, CRCL still achieves competitive results on well-labeled datasets, as shown in the following table:
> ||I2T|I2T|I2T|T2I|T2I|T2I||
> |-|-|-|-|-|-|-|-
> ||R@1|R@5|R@10|R@1|R@5|R@10|rSum
> |CHAN|79.7|96.7|98.7|63.8|90.4|95.8|525.0
> |HREM|81.2|96.5|98.9|63.7|90.7|96.0|527.1
> |CRCL|80.7|96.5|98.6|65.1|91.2|96.1|528.2
>
> The table reports the results of I2T and T2I retrieval tasks on the MS-COCO 5 fold-1K test set.
>
> >[A] Noisy Correspondence Learning With Meta Similarity Correction, CVPR'23.\
> [B] Learning Semantic Relationship Among Instances for Image-Text Matching, CVPR'23.\
> [C] Fine-Grained Image-Text Matching by Cross-Modal Hard Aligning Network, CVPR'23.

---

### Official Review · Reviewer_8spS · 2023-07-05

**Soundness:** 3 good
**Presentation:** 2 fair
**Contribution:** 3 good
**Rating:** 6
**Confidence:** 4

**Summary:**

This manuscript focuses on image-text matching under the noisy correspondence setting. To achieve a noise robust multi-modal representation, the authors propose two components, including a Active Complementary Loss (ACL) and a Self-Refining Correspondence Correction (SRCC). In ACL, a complementary contrastive loss styled fomula is derived, in which a coefficient $q$ is set in seeking for a tighter bound between the divergence between risk of training with noisy correspondence and ideal setting. As for SRCC, the labeled matching score is relaxed by momentum updating to alleviate the noise. Finally, the authors conduct extensive experiments on image-text retrieval to show their performance.

**Strengths:**

1. The motivation of the manuscript is novel, which is from noise tolerance loss function designing and noisy label correction simultaneously.

2. The proposed Active Complementary Loss has rigorous theoretical proof in evidenting the tighter bound to the divergence between noisy risk with ideal risk.

3. The experiments are adequate and extensive. What' s more, as the Tab. 1 shown, the proposed method is very stable under different noisy ratios, and the improvment is non-trivial under larger noisy ratio.

**Weaknesses:**

1. Lacks necessary explaination in figures. Actually, the pure text claim to the proposed method could be harder to grab.

2. The Sec. 2.2 is complex and hard to follow. Beyond theoretical proof, how ACL works in intuition should be further discussed.

3. The existing organization to the proposed two components is poor. Actually, I didn' t see a clear connection between Sec. 2.2 with Sec. 2.3. Despite that they are all for noise correspondence, the author fail to discuss the relationship between ACL and SRCC. If not, I could treat this manuscript as a naive A+B technic.

**Questions:**

None

---

> ### Author Rebuttal · Authors · 2023-08-09
>
> Thanks for your valuable comments and constructive suggestions. We will address your concerns and questions one by one as follows.
>
> **Q1. Lacks necessary explanation in figures. Actually, the pure text claim to the proposed method could be harder to grab.**
>
> Thank you for your constructive comments and suggestion. To make our method more clear and more understandable, we have added a ***pdf file*** to ***the global Author Rebuttal*** that contains an illustration of our framework (Figure 1). Here is a brief explanation of the illustration, as follows:
>
> *This is an illustration of our CRCL framework. CRCL consists of two components: an Active Complementary Loss (ACL) and a Self-refining Correspondence Correction (SCC). ACL balances between active and complementary learning based on the matching probabilities computed by the cross-modal model. It uses the corrected labels from SCC to conduct reasonable direct cross-modal learning by focusing on the active loss of convincing data and also recasting the regulatory factor to provide robust tolerance for possible noisy data under risk minimization theory. SCC exploits multiple self-refining processes with momentum correction to enlarge the receptive field for correcting correspondences, thus obtaining accurate labels and alleviating error accumulation.*
>
>
> **Q2. The Sec. 2.2 is complex and hard to follow. Beyond theoretical proof, how ACL works in intuition should be further discussed.**
>
> Thank you for your valuable suggestion. The main idea behind our ACL is to balance between the underfitting property of complementary loss and the noise overfitting disadvantage of active loss, thus improving the robustness against noisy correspondence (NC). Previous works explore different robust losses to alleviate the adverse impact of NC, but they often face overfitting or underfitting problems, especially under high noise rates. For example, active losses can fit the data quickly but they will overfit the noise easily. On the other hand, some robust losses can resist the noise well but they will underfit the clean data. Therefore, we propose our ACL to achieve a balance between efficient learning and robustness. ACL introduces exponential normalization into complementary loss to control the risk difference between noisy and ideal clean data, thus ensuring robustness for noisy data. However, since complementary learning has a long-standing underfitting problem, we still need to pay more attention to positive/matched pairs for direct efficient learning. Therefore, we introduce a weighted active learning loss into ACL that focuses more on convincing pairs. We will give more intuitive explanations in the next version.
>
> **Q3. The existing organization of the proposed two components is poor. Actually, I didn't see a clear connection between Sec. 2.2 with Sec. 2.3. Despite that they are all for noise correspondence, the author fails to discuss the relationship between ACL and SCC. If not, I could treat this manuscript as a naive A+B technic.**
>
> Thank you for your detailed review. We would like to clarify that our proposed method is not a simple A+B technique. ACL and SCC are complementary and indispensable components in our proposed framework, which can also be seen from our ablation experiments in Table 4, i.e., SCC without our ACL will yield an inferior performance. More specifically, ACL actually needs to use corrected labels to provide an ideal weighted score for active loss and an ideal regulatory factor for robust complementary loss, which suggests that our ACL actually relies on a very reliable correspondence correction technique. SCC plays such a role and is essential for ACL. To obtain accurate weighted scores, SCC leverages Momentum Correction to aggregate historical predictions, providing stable and accurate correspondence estimations. Furthermore, SCC combines multiple independent self-refining processes in training to eliminate error accumulation against noisy correspondences. The elimination of error accumulation allows ACL to provide better learning objectives for the model, thereby improving performance. So, SCC and ACL are mutually beneficial in our framework actually.
>
> Thank you for your time and feedback. We hope this helps you understand our work better. If you have any more questions or comments, please let us know.

---

### Official Review · Reviewer_J1KY · 2023-07-07

**Soundness:** 3 good
**Presentation:** 2 fair
**Contribution:** 3 good
**Rating:** 5
**Confidence:** 4

**Summary:**

This paper focuses on the problem of noise correspondence in image-text matching tasks. To address this issue, this paper proposes a generalized cross-modal robust complementary learning framework, which not only reduces the risk of erroneous supervision from the active complementary loss but also obtains stable and accurate correspondence correction through a self-refining correspondence correction. Extensive experiments show that the developed model could significantly improve the effectiveness and robustness compared with the state-of-the-art approaches. The topic of this paper is of great practical interest and the motivation is clear.

**Strengths:**

a.	This paper proposes a novel generalized cross-modal robust complementary learning framework to address the noisy correspondence problem in image-text matching tasks, which enhances the effectiveness of existing methods through robust loss and correction techniques.
b.	This paper utilizes the active complementary loss that employs complementary pairs to conduct indirect cross-modal learning with exponential normalization to boost robustness against noisy correspondence. In addition, self-refining correspondence correction is proposed to obtain stable and accurate correspondence correction.
c.	Extensive experiments are provided to prove the effectiveness and robustness of the proposed model.


**Weaknesses:**

a.	Essentially, the authors propose a new loss function to solve the noise correspondence problem in the image-text matching task. The proposed active complementary loss and self-refining correspondence correction both serve the final loss. So I think the novelty may be limited.
b.	I noticed in the supplementary that when the framework proposed in this paper is applied to the VSE model, there is a leap in performance improvement (such as 60% noise, Flickr30K dataset, The bidirectional retrieval improved by 50.3% and 35.4% on R@1, respectively.) while the performance improvement was limited when applied to the SGRAF model. Therefore, I think the author should explain why the performance gap is so large. More extensive experiments should also be organized to demonstrate the robustness of the proposed framework.
c.	To my knowledge, in previous studies, the noise injected on the Flickr30K and MS COCO datasets was different. So how is the noise injected by the author generated?


**Questions:**

Please refer to the item "weaknesses".

**Limitations:**

In addition to the effectiveness of the experimental results, I suggest that the authors can apply the proposed framework in more methods to enhance its robustness.

---

> ### Author Rebuttal · Authors · 2023-08-09
>
> Thanks for your valuable comments. We have carefully looked into all comments. Attached is our point-by-point response.
>
> **Q1. Regarding the concern of novelty .**
>
> Thanks for your comment but we disagree with your opinion. Although some methods are proposed to address noisy correspondence (NC), they still face some outstanding issues that limit their performance: overfitting, underfitting, and inefficiency. To be specific, **(1)** Previous works explore different robust losses to alleviate the adverse impact of NC, however, almost all of them face overfitting or underfitting problems, especially under high noise rates. Specifically, active losses own fast-fitting ability but easily overfit noise. On the other hand, some robust losses have high robustness but easily underfit clean data. It is still an open issue of how to balance efficient learning and robustness, however, which is less touched so far. To address this issue, we present a dynamic balance strategy to make active and robust losses promote each other for robust learning. **(2)** Due to the memorization effect of DNNs, cross-modal learning will face the self-sample-selection error accumulation problem, which will degrade the performance. To alleviate the error accumulation, previous robust frameworks (e.g., NCR and BiCro) utilize the co-teaching manner to obtain accurate predictions. However, the co-teaching strategy would increase the number of models,  which will largely increase the training overhead. To address this problem, we present an efficient Self-refining Correspondence Correction (SCC) without any additional model to tackle the error accumulation problem. The efficiency comparison is shown in the response (Q.2) to Reviewer DZ4B.
>
> Furthermore, our method is a generalized framework that can endow existing image-text matching methods with robustness and stability under NC. We have extended several classical image-text matching methods, e.g., VSE$\infty$, SGR, SAF, and CLIP (see Q.2), all of which have shown superior performance and robustness. In summary, we think that this work is with sufficient novelty and technical contribution.
>
> **Q2. (a) I think the author should explain why the performance gap is so large. (CRCL-VSE$\infty$/SGRAF vs. VSE$\infty$/SGRAF) (b) More extensive experiments should also be organized to demonstrate the robustness of the proposed framework.**
>
> Thank you for your careful comment and valuable suggestion. **(a)** We think that the discrepancy in performance improvement is due to the differences in the network models. Different network models have different behaviors and sensitivities to noisy correspondences, which result in different absolute performance gains. For example, as shown in Table 2, the performance of VSE$\infty$ and the SGRAF models varies under different noise rates. Even for SGR and SAF, which have the same framework with a minor difference in a subnetwork, their performance differs significantly in Table 2. Therefore, the difference in network models affects how the methods perform under NC. This can also lead to a gap in absolute performance between different methods (e.g., VSE$\infty$, SAF, and SGR). Moreover, from the experimental results, we can see that our CRCL consistently improves the methods with different network models, which demonstrates the effectiveness and generalization of our method.
>
> **(b)** As suggested, we applied CRCL on the classic pre-trained model CLIP(ViT/32) to further verify the robustness of our method, in addition to the CRCL-VSE$\infty$, CRCL-SAF, and CRCL-SGR models reported in the manuscript. We conducted a more extensive evaluation on the Flickr30K dataset and obtained the following partial results:
> |||I2T|I2T|I2T|T2I|T2I|T2I|||
> |-|-|-|-|-|-|-|-|-|-
> |Configurations|Methods|R@1|R@5|R@10|R@1|R@5|R@10|rSum|$\uparrow$(%)
> |Zero-shot|CLIP|78.6|95.4|97.7|59.8|85.1|90.9|507.5
> |Finetune on 40% Noise|CLIP$_{best}$|81.9|95.3|98.1|66.1|88.7|93.6|523.7
> |Finetune on 40% Noise|CLIP$_{last}$|64.7|86.8|91.1|45.7|68.5|76.2|433.0|-90.7%
> |Finetune on 40% Noise|CRCL-CLIP$_{best}$|83.2|96.2|98.7|67.3|89.5|94.2|529.1
> |Finetune on 40% Noise|CRCL-CLIP$_{last}$|83.2|96.2|98.6|67.4|89.5|94.2|529.1|-0.0%
> |Finetune on 80% Noise|CLIP$_{best}$|71.0|90.0|95.0|54.5|79.7|87.1|477.3
> |Finetune on 80% Noise|CLIP$_{last}$|28.8|51.4|63.0|17.5|33.9|42.2|236.8|-240.5%
> |Finetune on 80% Noise|CRCL-CLIP$_{best}$|83.0|96.1|98.4|66.8|89.1|94.0|527.4
> |Finetune on 80% Noise|CRCL-CLIP$_{last}$|83.1|96.0|98.68|66.7|89.1|94.0|527.5|+0.1%
>
> Note that $best$ means the results from testing the checkpoint with the best validation performance, and $last$ means using the last checkpoint to perform testing. By comparing the results of the $best$ and $last$ rows, we can reflect the degree of overfitting NC by the performance degradation, and then understand the robustness of our method. From the results, our CRCL not only improves the performance of CLIP on the dataset with NC, but also alleviates the performance degradation (compare $best$ and $last$ rows) caused by noise overfitting, which demonstrates the strong robustness of CRCL. For more verification of the robustness, we can also refer to Fig. 2 in the supplementary material. Fig. 2 records the variation in validation performance during training. Except for the full version of CRCL, the rest variants suffer from a noise overfitting problem, such as the CCL loss, $\mathcal{L}_d$, and the TR loss, which also shows that CRCL has better robustness.
>
> **Q3. Regarding the method to generate noisy correspondence.**
>
> Thank you for your detailed comments. To generate noisy correspondence, we followed the standard method used in NCR [12], which is **randomly shuffling the captions of training images for a specific percentage**. This way, we can ensure a fair comparison with previous works by using the same noise. For a more comprehensive evaluation, we reproduced the results of some baselines under four noise rates.

---

### Author Rebuttal · Authors · 2023-08-09

This is a global response. We add the illustration of our method in the attached pdf file.

---

### Decision · Program_Chairs · 2023-09-21

**Decision:**

Accept (poster)

**Comment:**

This paper focuses on cross-modal active complementary learning with self-refining correspondence. It receives all five positive ratings. It seems that all reviewers are satisfied with the writing, interesting idea, theoretical analysis, and experimental results. For the concerns, the authors also present detailed responses. Overall, this is a good paper, and I vote for its acceptance.